# Recyclable and Stable *α*-Methylproline-Derived Chiral Ligands for the Chemical Dynamic Kinetic Resolution of free *C,N*-Unprotected *α*-Amino Acids

**DOI:** 10.3390/molecules24122218

**Published:** 2019-06-13

**Authors:** Shuangjie Shu, Liang Zhao, Shengbin Zhou, Chenglin Wu, Hong Liu, Jiang Wang

**Affiliations:** State Key Laboratory of Drug Research, Shanghai Institute of Materia Medica, Chinese Academy of Sciences, 555 Zu Chong Zhi Road, Shanghai 201203, China; shushuangjie@126.com (S.S.); frankzl@163.com (L.Z.); stbin_06@163.com (S.Z.); wucl1990@mail.ustc.edu.cn (C.W.)

**Keywords:** quaternary carbon, dynamic kinetic resolution, unprotected *α*-amino acids, Schiff base intermediates, ligands

## Abstract

A novel special designed, stable, and recyclable chiral ligand bearing a quaternary carbon was developed for chemical dynamic kinetic resolution (DKR) of free *C,N*-unprotected racemic α-amino acids via Schiff base intermediates. This method furnishes high yields with excellent enantioselectivity, has a broad substrate scope, and uses operationally simple and convenient conditions. The present chemical DKR is a practical and useful method for the preparation of enantiopure α-amino acids.

## 1. Introduction

Unnatural *α*-amino acids play an important role in modern organic synthesis and medicinal chemistry due to their promising biological and pharmaceutical activity [1,2,3,4,5,6,7,8,9]. Considering the pharmaceutical potential of unnatural *α*-amino acids, the preparation of structurally varied and enantiomerically pure *α*-amino acids has been a focal point for synthetic chemists, thereby leading to many truly ingenious approaches. By using enantioselective polymer membranes, optical resolution of racemic *α*-amino acids to obtain enantiopure isomers has been established [10,11,12,13,14]. Meanwhile, several methods using chromatography were applied for the enantioselective separation of racemic *α*-amino acids and derivatives. Nevertheless, using enantioselective polymer membrane and chiral stationary phase are unarguably not economic, especially on a large scale [15,16,17].

Large-scale preparation of unnatural *α*-amino acids is virtually entirely accomplished by biocatalytic methods [18,19,20,21,22], and most enantioselective chemical methods are still undeveloped [23,24,25,26,27,28,29]. Among established chemical methods to enantioselectively synthesize pure *α*-amino acids, the dynamic kinetic resolution (DKR) of *α*-amino acids on large scale is the most efficient solution [30,31,32,33]. In 2012, Oh et al. [26]. Synthesized *α*-amino acids via DKR with limited substrate generality and using expensive thiourea organocatalysts starting from N-protected racemic amino acids. Tokumaru et al. [29]. Demonstrated DKR of racemic 2-(1*H*-pyrrol-1-yl)alkanoic acids as *α*-amino acid equivalents with sound yields and enantioselectivities, although the substrate scope was limited and the preparation of free *α*-amino acids was complicated (Scheme 1). Several research groups have developed practical methods for the preparation of chiral unnatural *α*-amino acids and *β*-amino acids [34,35,36,37,38,39,40,41]. In particular, chiral ligands **1**, **2**, and **3** (Scheme 1) were successfully used for the preparation of structurally varied *α*-amino acids, respectively [42,43,44,45,46]. In 2015, our group realized chemical dynamic thermodynamic resolution and *S*/*R* interconversion of unprotected unnatural *α*-amino acids [43,44]. Unfortunately, ligand **1** has limited substrate generality and incomplete stereochemical outcome, and both of the chiral ligands **1** and **2** were synthesized bearing axially chiral 1,1’-binaphthyl moieties, which were rather expensive [43]. Our group used recyclable ligand **3** for the dynamic kinetic resolution of *α*-amino acids [44]. We discovered that ligands bearing quaternary carbons could be used for chemical resolution of unprotected *α*-substituted-*β*-amino acids and *β*-substituted-*β*-amino acids without racemization of the *α*-position of the proline moiety [47,48]. Herein, we have designed and synthesized a novel stable ligand **4** (Scheme 1) for the resolution of free unprotected *α*-amino acids under operationally simple and convenient conditions. The entioselective pure resolution of phenyl substituted *α*-amino acids could also be realized directly and efficiently, thus expanding the scope of substrates, which holds great potential for preparation of enantiopure *α*-amino acids.

## 2. Results and Discussions

To investigate the most favorable reaction conditions, we optimized the solvents, bases, and temperature, using (*S*)-**4** and racemic phenylalanine (**5a**) as the template reaction (Table 1). Based on our previous study, DKR processes in protonic solvent may afford better yields and stereochemical control [40]. To our delight, excellent diastereoselectivity (dr > 99:1) was observed, both in ethanol using Ni(OAc)_2_ as Ni(II) ion resource and K_2_CO_3_ as base (Table 1, entry 1). Replacing the ethanol solvent with methanol led to a significant improvement in the yield (up to 98%) and also afforded product **6a** with excellent diastereoselectivity (>99:1, Table 1, Entry 2), whereas using sodium *tert*-butoxide (NaO*t*-Bu), sodium hydride (NaH), and 1,8-diazabicyclo[5.4.0] undec-7-ene (DBU) as the base, all led to a significant drop in the reaction yield (Table 1, Entries 3–5). The yield thus greatly depends on the conditions applied, while the stereochemical outcome does not appear to be influenced by the nature of both the base and solvent used. Next, the effect of temperature was explored, and a good yield with excellent diastereoselectivity (dr > 99:1, Table 1, Entry 6) could be obtained at room temperature within 72 h. All diastereoselectivities were monitored by LC-MS, and the (*S*,2*S*)-**6a** was obtained as the major product, which was determined by single-crystal X-ray diffraction analysis (Table 1, X-ray, see the Materials and Methods Section for details).

With optimized reaction conditions in hand, we further investigated the scope of substrate generality (Table 2, dr determination, see Appendix A). In all cases the stereochemistry outcome showed that all (*S*)-**4** starting materials gave enantiomerically enriched compounds (*S*,2*S*)-**6** as the major DKR products, respectively. We explored several phenylalanine derivatives. Simple substituted phenylalanines proved to be good substrates (Table 2, Entries 2–5), providing the desired products in high yields (88-91%) with excellent diastereoselectivity (dr > 99:1). The reactions of (*S*)-**4** with *α*-amino acids containing functionalized phenylalanines (Table 2, Entry 6), fused aryl rings, and heterocyclic aryl rings (Table 2, Entries 7–9) were conducted and high yields (89–94%) were obtained with excellent stereochemical outcomes (dr > 99:1). We also investigate the phenyl-substituted glycine (Table 2, Entries 10 and 11). Methoxy- and bromo-substitution on the *para*-position of phenyl both gave good yields and diastereoselectivity outcomes. Furthermore, the reaction could be extended to alkyl substituted *α*-amino acids (Table 2, Entries 12–17). All these reactions produced high yields (87–95%) with excellent diastereoselectivity (dr 97:3 to >99:1).

After successful exploration of (*S*)-**4**, we turned our attention to the (*R*)-**4** ligand (Table 3, dr determination, see Appendix A). We found that Ni(II)-complexes (*R*,2*R*)-**6** were synthesized in high yields (70–98%) with excellent diastereoselectivity (dr 97:3 to > 99:1). As proved by (*S*)-**4**, the resolution of phenylalanine and substituted phenylalanines (Table 3, Entries 1-6) was realized in high yields (88–91%) with excellent diastereoselectivity (>99:1). Apart from that, *α*-amino acids with fused rings and heterocyclic aryl rings (Table 3, Entries 7–9) all exhibited both excellent yields (92–94%) and diastereoselectivity (>99:1). The resolution of phenyl-substituted glycines (Table 3, Entries 10 and 11) by (*R*)-**4** produced good yields and diastereoselective outcomes (97:3 and 98:2). All alkyl substituted *α*-amino acids (Table 3, Entries 12–17) obtained high yields (88–96%) with excellent diastereoselectivity (dr 98:2 to >99:1). With these remarkable results, we can offer a potential method to prepare unprotected (*R*)-*α*-amino acids with broad generality.

Compared to reported methods [43,44], we realized good to excellent yields an diastereoselectivities for phenyl-substituted glycine. We extended the substrate scope to include compounds such as cyclobutyl- and trifluoroethyl-substituted glycine, again with excellent yields and diastereoselectivities. Furthermore, the novel chiral ligand is more stable than the previous ligand.

With the aim of understanding the DKR progress of this reaction, LC-MS analysis was used to monitor the stereochemical outcome over time (Figure 1). The DKR reaction was conducted using ligand (*R*)-**4**, 2-(3-methoxyphenyl)glycine (**5j**), Ni(OAc)_2_, and K_2_CO_3_ as base at 60 °C in methanol. The ligand (*R*)-**4** readily reacted with (*R*)-**5j** in the initial stages of the process, thus giving rise the products (*R,*2*R*)-**6j** and (*R*,2*S*)-**6j** in 50% yield and a dr of 98:2 at 4 h. After 8 h, the yields of product increased steadily at a slow rate. However, the ratio between two diastereomers (*R*,2*R*)-**6j** and (*R*,2*S*)-**6j** is almost the same as after 4 h, which indicated that any base-catalyzed epimerization from (*R*,2*R*)-**6j** and (*R*,2*S*)-**6j** occured at a relatively slow rate. Considering the yield is almost consistent, final thermodynamic control was stopped at 16 h. The calculated yield of product (*R*,2*R*)-**6j**, as a single diastereomer, was 70% more than theoretical percentage of 50%. Herein, we assume that the disfavored *α*-amino acid (*S*)-**5j** undergoes base-catalyzed *α*-epimerization before forming the major product (*R*,2*R*)-**6j**.

Finally, decomposition of Ni(II) complex (*S*,2*S*)-**6a** under standard conditions afforded the target amino acid (*S*)-configured phenylalanine ((*S*)-**5a**) in high yield (88%) with excellent enantioselectivity (ee > 99%). The stable chiral ligand (*S*)-**4** could be recycled and reused in high yield (90%) (Figure 2, ee determination see Appendix A).

## 3. Materials and Methods 

### 3.1. General Information

All commercially available compounds and solvents were used without further purification. Anhydrous nickel acetate was prepared by heating Ni(II) acetate tetrahydrate at 110 °C under vacuum for 2 h. ^1^H- and ^13^C-NMR spectra were recorded in deuterochloroform (CDCl_3_) on a 400 or 500 MHz AV400 instrument (Bruker, Billerica, MA, USA) (see Appendix A). Chemical shifts were reported in parts per million (ppm, *δ*) downfield from tetramethylsilane. Proton coupling patterns were described as singlet (s), doublet (d), triplet (t), quartet (q), broad (br) and multiplet (m). High resolution mass spectra (HRMS) were measured on a Finnigan MAT95 spectrometer (Thermo Fisher Scientific, Waltham, MA, USA) The determination of dr was performed via LC-MS analysis (Finnigan MAT95, Thermo Fisher Scientific). The determination of ee was performed via HPLC analysis (Agilent-1100, Santa Clara, CA, USA). Optical rotations were measured using a 1 mL cell with a 10 cm path length on an Autopol VI automatic polarimeter (Rudolph Research Analytical, Hackettstown, NJ, USA) and are reported as follows: [*α*]^20^_D_ (c: g/100 mL, in solvent). Melting points were measured on a melting point apparatus (SGW-X4; Shanghai ZiQi Laboratory Equipment Co., Ltd., Shanghai, China).

### 3.2. General Procedure for the Synthesis of (S,2S)-***6a*** and (R,2R)-***6a***

A suspension of (*S*)-**4 or** (*R*)-**4** (100.3 mg, 0.2 mmol, 1 equiv.), *rac*-phenylalanine **5a** (33 mg, 0.2 mmol, 1 equiv.), Ni(OAc)_2_ (35.3 mg, 0.2 mmol, 1 equiv.), and K_2_CO_3_ (138.1 mg, 1.0 mmol, 5 equiv.) were refluxed in methanol (4 mL) at 60 °C for 8 h. After cooling to room temperature, the mixture was diluted with 5% aqueous acetic acid (15 mL) and extracted three times with dichloromethane. The combined organic layers were dried over Na_2_SO_4_ and concentrated under reduced pressure. The crude mixture was purified by column chromatography on silica gel (PE/EA = 4:1 to DCM/MeOH = 20:1) to afford two diastereomers (*S*,2*S*)-**6a** and (*S*,2*R*)-**6a** (138 mg, yield 98%) for analysis (dr > 99:1). The mixture was purified again by column chromatography on silica gel (DCM/MeOH = 40:1) to give the pure diastereomer (*S*,2*S*)-**6a** as a red solid.

### 3.3. General Procedure for the Synthesis of (S)-***5a***

To a suspension of (*S*,2*S*)-**6a** (78 mg, 0.1 mmol, 1 equiv.) in 1,4-dioxane (2.2 mL) was added 6 *N* HCl (618 μL) and the mixture was then heated at 70 °C for 10 min. The reaction was monitored by TLC and disappearance of the red color of the solution was observed. After concentrating to dryness under reduced pressure, ethyl acetate and water were added to the residue and then stirred. The organic phase was extracted two times with water and washed with saturated sodium hydrogen carbonate solution, water, brine, and finally dried over Na_2_SO_4_, filtered and concentrated to produce recovered (*S*)-**4** (50.8 mg, 90%). The combined aqueous phase was concentrated in vacuo and dissolved in ammonium hydroxide solution (10%), which subsequently loaded onto a Dowex 50WX2-100 cation-exchange resin column (Sigma-Aldrich, St. Louis, MI, USA). The column was preprocessed with deionized water until neutral, and then obtained the desired amino acid using ammonium hydroxide (10%). The aqueous solution was concentrated in vacuo to give (*S*)-**5a** (16.5 mg, 88%) as a white solid.

### 3.4. Analytical Characterization Data of Products

*Nickel(II)-(S)-N-(2-benzoyl-4-chlorophenyl)-1-(3,4-dichlorobenzyl)-2-methylpyrrolidine-2-carboxamide/(S)-phenylalanine Schiff Base Complex* (**6a**). Red solid (138 mg, yield 98%); mp 116–118 °C; [*α*]^20^_D_ = +2502 (c = 1, CHCl_3_); ^1^H-NMR (400 MHz, CDCl_3_) *δ* 9.08 (d, *J* = 2.1 Hz, 1H), 8.32 (d, *J* = 9.3 Hz, 1H), 7.61–7.49 (m, 3H), 7.42–7.27 (m, 6H), 7.15–7.02 (m, 3H), 6.65 (d, *J* = 7.7 Hz, 1H), 6.61 (d, *J* = 2.6 Hz, 1H), 4.22 (t, *J* = 5.2 Hz, 1H), 3.54 (dd, *J* = 59.9, 13.0 Hz, 2H), 3.36–3.24 (m, 1H), 3.14 (dt, *J* = 17.6, 6.8 Hz, 2H), 2.78 (dd, *J* = 13.8, 5.4 Hz, 1H), 2.42–2.23 (m, 1H), 2.14–2.03 (m, 1H), 1.98 (dd, *J* = 19.7, 9.7 Hz, 1H), 1.87–1.73 (m, 1H), 1.33 (s, 3H). ^13^C-NMR (125 MHz, CDCl_3_) *δ* 180.5, 178.1, 170.8, 141.5, 135.8, 134.0, 133.4, 133.1, 133.0, 132.7, 132.6, 131.1, 130.5, 130.3, 129.8, 129.5, 129.3, 129.1, 127.9, 127.7, 127.5, 125.8, 124.3, 74.3, 71.8, 57.3, 54.7, 41.4, 39.8, 20.7, 17.8. LRMS (ESI, *m*/*z*): 704.0 [M + H]^+^. HRMS (ESI, *m*/*z*): calcd for C_35_H_31_Cl_3_N_3_NiO_3_, 704.0605 [M + H]^+^; found 704.0625. The dr was determined by LC-MS using a system equipped with a binary pump, and a photodiode array detector (DAD), using an Eclipse XDB-C18 column (250 × 4.6 mm, 5 µm) (CH_3_CN/H_2_O = 65:35, flow rate 1.0 mL/min, λ = 254 nm), t_major_ = 15.966 min, t_minor_ = not found, dr > 99:1.

*Nickel(II)-(S)-N-(2-benzoyl-4-chlorophenyl)-1-(3,4-dichlorobenzyl)-2-methylpyrrolidine-2-carboxamide/(S)-2-methoxyphenylalanine Schiff Base Complex* (**6b**). Red solid (132 mg, yield 90%); mp 125–128 °C; [*α*]^20^_D_ = +2021 (c = 1, CHCl_3_); ^1^H-NMR (400 MHz, CDCl_3_) *δ* 9.00 (d, *J* = 2.0 Hz, 1H), 8.36 (d, *J* = 9.3 Hz, 1H), 7.57–7.53 (m, 2H), 7.52–7.48 (m, 1H), 7.41–7.37 (m, 1H), 7.36–7.31 (m, 1H), 7.31–7.27 (m, 2H), 7.22–7.19 (m, 1H), 7.10 (dd, *J* = 9.3, 2.6 Hz, 1H), 6.96 (t, *J* = 7.4 Hz, 1H), 6.88–6.82 (m, 2H), 6.62 (d, *J* = 2.6 Hz, 1H), 4.15 (t, *J* = 4.9 Hz, 1H), 3.52 (q, *J* = 13.4 Hz, 2H), 3.31–3.22 (m, 4H), 3.21–3.16 (m, 1H), 3.11–3.06 (m, 1H), 2.91 (dd, *J* = 13.6, 4.9 Hz, 1H), 2.31–2.20 (m, 1H), 2.10–2.01 (m, 1H), 1.94 (q, *J* = 9.9 Hz, 1H), 1.83–1.73 (m, 1H), 1.31 (s, 3H). ^13^C-NMR (125 MHz, CDCl_3_) *δ* 180.5, 178.6, 171.1, 158.5, 141.4, 135.9, 134.1, 133.7, 133.2, 133.1, 132.9, 132.7, 132.5, 131.2, 130.1, 130.0, 129.4, 129.3, 129.3, 128.4, 128.0, 127.7, 125.7, 124.5, 124.2, 121.5, 110.7, 74.6, 72.0, 57.7, 54.9, 54.6, 41.4, 34.8, 20.9, 18.1. LRMS (ESI, *m*/*z*): 734.0 [M − H]^−^. HRMS (ESI, *m*/*z*): calcd for C_36_H_31_Cl_3_N_3_NiO_4_, 734.0739 [M − H]^−^; found 734.0741. The dr was determined by LC-MS using an Eclipse XDB-C18 column (250 × 4.6 mm, 5 µm) (CH_3_CN/H_2_O = 65:35, flow rate 1.0 mL/min, λ = 254 nm), t_major_ = 19.059 min, t_minor_ = not found, dr > 99:1.

*Nickel(II)-(S)-N-(2-benzoyl-4-chlorophenyl)-1-(3,4-dichlorobenzyl)-2-methylpyrrolidine-2-carboxamide/(S)-3-methoxyphenylalanine Schiff Base Complex* (**6c**). Red solid (127 mg, yield 90%); mp 118–121 °C; [*α*]^20^_D_ = +3344 (c = 1, CHCl_3_); ^1^H-NMR (400 MHz, CDCl_3_) *δ* 9.07–9.05 (m, 1H), 8.30 (d, *J* = 9.3 Hz, 1H), 7.59–7.45 (m, 3H), 7.36–7.27 (m, 3H), 7.23–7.18 (m, 1H), 7.09 (dd, *J* = 9.3, 2.6 Hz, 1H), 6.88–6.83 (m, 1H), 6.64 (d, *J* = 7.5 Hz, 1H), 6.59–6.56 (m, 2H), 6.55–6.50 (m, 1H), 4.19 (t, *J* = 5.3 Hz, 1H), 3.63 (s, 3H), 3.60 (d, *J* = 13.1 Hz, 1H), 3.45 (d, *J* = 13.2 Hz, 1H), 3.42–3.34 (m, 1H), 3.22–3.09 (m, 2H), 2.78 (dd, *J* = 13.8, 5.1 Hz, 1H), 2.49–2.36 (m, 1H), 2.17–2.08 (m, 1H), 2.04–1.94 (m, 1H), 1.91–1.79 (m, 1H), 1.33 (s, 3H). ^13^C-NMR (125 MHz, CDCl_3_) *δ* 180.8, 178.3, 171.0, 141.6, 135.9, 135.8, 134.1, 133.5, 133.2, 133.1, 132.8, 132.7, 131.2, 130.4, 129.9, 129.7, 129.6, 129.4, 127.8, 127.5, 126.5, 125.9, 124.4, 124.4, 74.5, 71.4, 57.5, 54.9, 54.6, 41.6, 34.2, 21.0, 18.0. LRMS (ESI, *m*/*z*): 734.0 [M − H]^−^. HRMS (ESI, *m*/*z*): calcd for C_36_H_31_Cl_3_N_3_NiO_4_, 734.0739 [M − H]^−^; found 734.0741. The dr was determined by LC-MS using an Eclipse XDB-C18 column (250 × 4.6 mm, 5 µm) (CH_3_CN/H_2_O = 65:35, flow rate 1.0 mL/min, λ = 254 nm), t_major_ = 16.298 min, t_minor_ = not found, dr > 99:1.

*Nickel(II)-(S)-N-(2-benzoyl-4-chlorophenyl)-1-(3,4-dichlorobenzyl)-2-methylpyrrolidine-2-carboxamide/(S)-3-methylphenylalanine Schiff Base Complex* (**6d**). Red solid (126 mg, yield 88%); mp 135–136 °C; [*α*]^20^_D_ = +2588 (c = 1, CHCl_3_); ^1^H-NMR (400 MHz, CDCl_3_) *δ* 9.06–9.03 (m, 1H), 8.33 (d, *J* = 9.3 Hz, 1H), 7.56 (t, *J* = 7.7 Hz, 2H), 7.51 (t, *J* = 7.5 Hz, 1H), 7.37 (t, *J* = 7.5 Hz, 1H), 7.33–7.27 (m, 2H), 7.24–7.18 (m, 1H), 7.16–7.09 (m, 2H), 6.89–6.83 (m, 2H), 6.64 (d, *J* = 7.6 Hz, 1H), 6.61 (d, *J* = 2.5 Hz, 1H), 4.20 (t, *J* = 5.1 Hz, 1H), 3.63–3.55 (m, 1H), 3.50–3.42 (m, 1H), 3.35–3.24 (m, 1H), 3.16–3.06 (m, 2H), 2.74 (dd, *J* = 13.7, 5.4 Hz, 1H), 2.41–2.29 (m, 1H), 2.26 (s, 3H), 2.15–2.03 (m, 1H), 2.02–1.94 (m, 1H), 1.88–1.76 (m, 1H), 1.32 (s, 3H). ^13^C-NMR (150 MHz, CDCl_3_) *δ* 180.2, 178.0, 170.5, 141.3, 138.5, 135.6, 135.4, 133.8, 133.1, 132.9, 132.8, 132.4, 132.4, 131.1, 131.1, 130.9, 130.0, 129.6, 129.3, 128.9, 128.7, 128.2, 127.7, 127.5, 127.3, 127.3, 125.5, 124.0, 74.1, 71.6, 57.1, 54.5, 41.0, 39.6, 21.4, 20.4, 17.6. LRMS (ESI, *m*/*z*): 716.0 [M − H]^−^. HRMS (ESI, *m*/*z*): calcd for C_36_H_31_Cl_3_N_3_NiO_3_, 716.0790 [M − H]^−^; found 716.0798. The dr was determined by LC-MS using an Eclipse XDB-C18 column (250 × 4.6 mm, 5 µm) (CH_3_CN/H_2_O = 65:35, flow rate 1.0 mL/min, λ = 254 nm), t_major_ = 21.599 min, t_minor_ = not found, dr > 99:1.

*Nickel(II)-(S)-N-(2-benzoyl-4-chlorophenyl)-1-(3,4-dichlorobenzyl)-2-methylpyrrolidine-2-carboxamide/(S)-4-fluorophenylalanine Schiff Base Complex* (**6e**). Red solid (132 mg, yield 91%); mp 188–190 °C; [*α*]^20^_D_ = +1786 (c = 1, CHCl_3_); ^1^H-NMR (400 MHz, CDCl_3_) *δ* 9.09–9.06 (m, 1H), 8.33 (d, *J* = 9.3 Hz, 1H), 7.62–7.52 (m, 3H), 7.47–7.41 (m, 1H), 7.34 (d, *J* = 7.0 Hz, 1H), 7.31 (d, *J* = 8.2 Hz, 1H), 7.14 (dd, *J* = 9.3, 2.5 Hz, 1H), 7.08–7.03 (m, 4H), 6.74 (d, *J* = 7.5 Hz, 1H), 6.63 (d, *J* = 2.5 Hz, 1H), 4.21 (t, *J* = 5.0 Hz, 1H), 3.64–3.58 (m, 1H), 3.52–3.46 (m, 1H), 3.40–3.30 (m, 1H), 3.17–3.03 (m, 2H), 2.75 (dd, *J* = 14.0, 5.4 Hz, 1H), 2.42–2.30 (m, 1H), 2.23–2.14 (m, 1H), 2.06–1.96 (m, 1H), 1.90–1.84 (m, 1H), 1.35 (s, 3H). ^13^C-NMR (150 MHz, CDCl_3_) *δ* 180.3, 177.7, 170.7, 162.3 (d, *J* = 246.68 Hz), 141.3, 135.5, 133.7, 133.1, 132.9, 132.8, 132.5, 132.4, 131.7, 131.7, 131.3, 131.3, 130.9, 130.2, 129.6, 129.4, 129.1, 127.5, 127.4, 127.3, 125.7, 124.1, 115.8, 115.7, 74.2, 71.4, 57.2, 54.4, 41.0, 38.6, 20.5, 17.6. LRMS (ESI, *m*/*z*): 720.0 [M − H]^−^. HRMS (ESI, *m*/*z*): calcd for C_35_H_28_Cl_3_FN_3_NiO_3_, 720.0539 [M − H]^−^; found 720.0547. The dr was determined by LC-MS using an Eclipse XDB-C18 column (250 × 4.6 mm, 5 µm) (CH_3_CN/H_2_O = 65:35, flow rate 1.0 mL/min, λ = 254 nm), t_major_ = 17.940 min, t_minor_ = not found, dr > 99:1.

*Nickel(II)-(S)-N-(2-benzoyl-4-chlorophenyl)-1-(3,4-dichlorobenzyl)-2-methylpyrrolidine-2-carboxamide/(S)-3,5-diiodotyrosine Schiff Base Complex* (**6f**). Red solid (175 mg, yield 90%); mp 286–288 °C; [*α*]^20^_D_ = +2037 (c = 1, CHCl_3_); ^1^H-NMR (400 MHz, CDCl_3_) *δ* 9.05 (d, *J* = 2.0 Hz, 1H), 8.43 (d, *J* = 9.3 Hz, 1H), 7.66–7.53 (m, 3H), 7.47 (dd, *J* = 10.8, 5.7 Hz, 1H), 7.33 (t, *J* = 7.2 Hz, 2H), 7.27 (s, 2H), 7.16 (dd, *J* = 9.3, 2.6 Hz, 1H), 6.78 (d, *J* = 7.4 Hz, 1H), 6.66 (d, *J* = 2.5 Hz, 1H), 5.82 (s, 1H), 4.14–4.07 (m, 1H), 3.56 (dd, *J* = 29.2, 13.0 Hz, 2H), 3.46–3.34 (m, 1H), 3.17 (t, *J* = 8.8 Hz, 1H), 2.89 (dd, *J* = 14.1, 4.4 Hz, 1H), 2.67 (dd, *J* = 13.9, 6.2 Hz, 1H), 2.54–2.39 (m, 1H), 2.24 (dd, *J* = 13.8, 9.8 Hz, 1H), 2.02 (td, *J* = 18.7, 9.4 Hz, 2H), 1.37 (s, 3H). ^13^C-NMR (125 MHz, CDCl_3_) *δ* 180.4, 177.6, 171.0, 153.4, 141.8, 140.6, 135.7, 134.0, 133.2, 133.1, 133.0, 132.6, 131.8, 131.2, 130.6, 129.8, 129.7, 129.4, 127.6, 127.5, 127.4, 125.8, 124.4, 82.8, 74.5, 71.2, 57.5, 54.7, 41.0, 38.0, 21.0, 17.9. LRMS (ESI, *m*/*z*): 971.8 [M + H]^+^. HRMS (ESI, *m*/*z*): calcd for C_35_H_29_Cl_3_I_2_N_3_NiO_4_, 971.8487 [M − H]^−^; found 971.8509. The dr was determined by LC-MS using an Eclipse XDB-C18 column (250 × 4.6 mm, 5 µm) (CH_3_CN/H_2_O = 65:35, flow rate 1.0 mL/min, λ = 254 nm), t_major_ = 17.421 min, t_minor_ = not found, dr > 99:1.

*Nickel(II)-(S)-N-(2-benzoyl-4-chlorophenyl)-1-(3,4-dichlorobenzyl)-2-methylpyrrolidine-2-carboxamide/(S)-3-(1-naphthyl)alanine Schiff Base Complex* (**6g**). Red solid (139 mg, yield 92%); mp 182–184 °C; [*α*]^20^_D_ = +2432 (c = 1, CHCl_3_); ^1^H-NMR (400 MHz, CDCl_3_) *δ* 9.12–9.09 (m, 1H), 8.31 (d, *J* = 9.3 Hz, 1H), 7.80 (t, *J* = 8.0 Hz, 2H), 7.69 (dd, *J* = 8.2, 2.0 Hz, 1H), 7.58 (d, *J* = 8.6 Hz, 1H), 7.42–7.37 (m, 1H), 7.36–7.28 (m, 4H), 7.21 (t, *J* = 7.7 Hz, 1H), 7.19–7.12 (m, 2H), 7.09 (dd, *J* = 9.3, 2.6 Hz, 1H), 6.69 (t, *J* = 7.6 Hz, 1H), 6.45 (d, *J* = 2.6 Hz, 1H), 5.65 (d, *J* = 7.7 Hz, 1H), 4.45–4.40 (m, 1H), 4.12–4.04 (m, 1H), 3.83–3.77 (m, 1H), 3.71 (d, *J* = 13.2 Hz, 1H), 3.67–3.57 (m, 1H), 3.47 (d, *J* = 13.2 Hz, 1H), 3.30–3.20 (m, 1H), 2.84–2.54 (m, 1H), 2.35–2.25 (m, 1H), 2.13 (q, *J* = 9.6 Hz, 1H), 2.07–1.95 (m, 1H), 1.41 (s, 3H). ^13^C-NMR (150 MHz, CDCl_3_) *δ* 180.1, 178.1, 170.3, 141.1, 135.6, 134.0, 133.6, 133.1, 132.7, 132.6, 132.4, 132.3, 132.1, 131.5, 131.0, 129.5, 129.4, 128.6, 128.5, 128.4, 128.3, 128.3, 127.5, 127.3, 127.2, 126.3, 125.8, 125.5, 125.4, 123.9, 123.2, 74.2, 71.4, 56.7, 54.9, 41.4, 40.0, 20.7, 17.7. LRMS (ESI, *m*/*z*): 752.0 [M − H]^−^. HRMS (ESI, *m*/*z*): calcd for C_39_H_31_Cl_3_N_3_NiO_3_, 752.0790 [M − H]^−^; found 752.0799. The dr was determined by LC-MS using an Eclipse XDB-C18 column (250 × 4.6 mm, 5 µm) (CH_3_CN/H_2_O = 65:35, flow rate 1.0 mL/min, λ = 254 nm), t_major_ = 24.156 min, t_minor_ = not found, dr > 99:1.

*Nickel(II)-(S)-N-(2-benzoyl-4-chlorophenyl)-1-(3,4-dichlorobenzyl)-2-methylpyrrolidine-2-carboxamide/(S)-3-(3-benzothienyl)alanine Schiff Base Complex* (**6h**). Red solid (142 mg, yield 94%); mp 138–141 °C; [*α*]^20^_D_ = +2452 (c = 1, CHCl_3_); ^1^H-NMR (400 MHz, CDCl_3_) *δ* 9.08–9.05 (m, 1H), 8.35 (d, *J* = 9.3 Hz, 1H), 7.87 (d, *J* = 8.0 Hz, 1H), 7.76–7.56 (m, 1H), 7.50 (t, *J* = 7.5 Hz, 1H), 7.43 (d, *J* = 8.0 Hz, 1H), 7.38 (t, *J* = 7.5 Hz, 1H), 7.34–7.28 (m, 3H), 7.19–7.10 (m, 4H), 6.60 (d, *J* = 2.6 Hz, 1H), 6.49 (d, *J* = 7.5 Hz, 1H), 4.32 (t, *J* = 5.5 Hz, 1H), 3.65–3.57 (m, 2H), 3.48–3.41 (m, 1H), 3.23–3.15 (m, 1H), 3.11–2.98 (m, 2H), 2.13–1.90 (m, 3H), 1.82–1.70 (m, 1H), 1.31 (s, 3H). ^13^C-NMR (125 MHz, CDCl_3_) *δ* 180.0, 178.1, 170.6, 141.3, 140.4, 139.1, 135.5, 133.7, 133.0, 132.8, 132.7, 132.5, 130.9, 130.3, 129.9, 129.6, 129.1, 128.8, 127.3, 127.3, 125.7, 125.5, 124.6, 124.3, 124.0, 122.7, 121.9, 74.1, 70.7, 57.0, 54.6, 40.8, 33.8, 20.2, 17.6. LRMS (ESI, *m*/*z*): 758.0 [M − H]^−^. HRMS (ESI, *m*/*z*): calcd for C_37_H_29_Cl_3_N_3_NiO_3_S, 758.0354 [M − H]^−^; found 758.0362. The dr was determined by LC-MS using an Eclipse XDB-C18 column (250 × 4.6 mm, 5 µm) (CH_3_CN/H_2_O = 65:35, flow rate 1.0 mL/min, λ = 254 nm), t_major_ = 27.103 min, t_minor_ = 24.123, dr > 99:1.

*Nickel(II)-(S)-N-(2-benzoyl-4-chlorophenyl)-1-(3,4-dichlorobenzyl)-2-methylpyrrolidine-2-carboxamide/(S)-3-(3-thienyl)alanine Schiff Base Complex* (**6i**). Red solid (126 mg, yield 89%); mp 178–180 °C; [*α*]^20^_D_ = +2012 (c = 1, CHCl_3_); ^1^H-NMR (400 MHz, CDCl_3_) *δ* 9.10 (d, *J* = 2.1 Hz, 1H), 8.35 (d, *J* = 9.3 Hz, 1H), 7.61–7.51 (m, 3H), 7.46–7.41 (m, 1H), 7.37–7.30 (m, 3H), 7.14 (dd, *J* = 9.3, 2.6 Hz, 1H), 7.06–7.04 (m, 1H), 6.86 (dd, *J* = 4.9, 1.2 Hz, 1H), 6.73–6.69 (m, 1H), 6.63 (d, *J* = 2.6 Hz, 1H), 4.18 (t, *J* = 5.1 Hz, 1H), 3.67–3.61 (m, 1H), 3.56–3.46 (m, 2H), 3.22–3.15 (m, 2H), 2.82–2.75 (m, 1H), 2.58–2.44 (m, 1H), 2.25–2.15 (m, 1H), 2.03 (q, *J* = 9.6 Hz, 1H), 1.97–1.87 (m, 1H), 1.37 (s, 3H). ^13^C-NMR (125 MHz, CDCl_3_) *δ* 180.4, 178.1, 170.7, 141.3, 135.6, 135.5, 133.8, 133.2, 132.9, 132.8, 132.5, 132.4, 130.9, 130.1, 129.6, 129.4, 129.3, 129.1, 127.5, 127.2, 126.2, 125.6, 124.1, 124.1, 74.2, 71.1, 57.2, 54.6, 41.3, 33.9, 20.7, 17.7. LRMS (ESI, *m*/*z*): 708.0 [M − H]^−^. HRMS (ESI, *m*/*z*): calcd for C_33_H_27_Cl_3_N_3_NiO_3_S, 708.0198 [M − H]^−^; found 708.0200. The dr was determined by LC-MS using an Eclipse XDB-C18 column (250 × 4.6 mm, 5 µm) (CH_3_CN/H_2_O = 65:35, flow rate 1.0 mL/min, λ = 254 nm), t_major_ = 15.723 min, t_minor_ = not found, dr > 99:1.

*Nickel(II)-(S)-N-(2-benzoyl-4-chlorophenyl)-1-(3,4-dichlorobenzyl)-2-methylpyrrolidine-2-carboxamide/(S)-2-(3-methoxyphenyl)glycine Schiff Base Complex* (**6j**). Red solid (124 mg, yield 87%); mp 153–155 °C; [*α*]^20^_D_ = +2183 (c = 1, CHCl_3_); ^1^H-NMR (400 MHz, CDCl_3_) *δ* 9.21–9.17 (m, 1H), 8.24 (d, *J* = 9.3 Hz, 1H), 7.82–7.78 (m, 1H), 7.58–7.51 (m, 1H), 7.44–7.36 (m, 3H), 7.30 (d, *J* = 7.5 Hz, 1H), 7.23 (t, *J* = 7.9 Hz, 1H), 7.16 (dd, *J* = 9.3, 2.5 Hz, 1H), 7.11–7.05 (m, 2H), 6.83–6.78 (m, 1H), 6.67 (d, *J* = 2.5 Hz, 1H), 6.15 (d, *J* = 7.9 Hz, 1H), 4.72 (s, 1H), 4.08–3.99 (m, 1H), 3.84–3.78 (m, 1H), 3.72 (s, 3H), 3.65–3.58 (m, 1H), 3.39–3.31 (m, 1H), 3.26–3.08 (m, 1H), 2.51–2.41 (m, 1H), 2.27–2.08 (m, 2H), 1.50 (s, 3H). ^13^C-NMR (150 MHz, CDCl_3_) *δ* 180.6, 177.2, 171.5, 159.6, 141.1, 139.0, 135.6, 133.8, 133.1, 133.1, 132.9, 132.4, 132.2, 131.0, 129.8, 129.7, 129.5, 128.8, 128.7, 127.5, 126.8, 126.5, 125.8, 124.6, 118.2, 74.5, 74.2, 57.0, 55.2, 55.0, 41.5, 21.0, 18.1. LRMS (ESI, *m*/*z*): 718.0 [M − H]^−^. HRMS (ESI, *m*/*z*): calcd for C_35_H_29_Cl_3_N_3_NiO_4_, 718.0583 [M − H]^−^; found 718.0602. The dr was determined by LC-MS using an Eclipse XDB-C18 column (250 × 4.6 mm, 5 µm) (CH_3_CN/H_2_O = 65:35, flow rate 1.0 mL/min, λ = 254 nm), t_major_ = 16.728 min, t_minor_ = 18.308, dr = 97:3.

*Nickel(II)-(S)-N-(2-benzoyl-4-chlorophenyl)-1-(3,4-dichlorobenzyl)-2-methylpyrrolidine-2-carboxamide/(S)-2-(3-bromophenyl)glycine Schiff Base Complex* (**6k**). Red solid (131 mg, yield 85%); mp 119–121 °C; [*α*]^20^_D_ = +2543 (c = 1, CHCl_3_); ^1^H-NMR (400 MHz, CDCl_3_) *δ* 9.20 (d, *J* = 2.1 Hz, 1H), 8.24 (d, *J* = 9.3 Hz, 1H), 8.17 (t, *J* = 1.9 Hz, 1H), 7.81 (dd, *J* = 8.2, 2.2 Hz, 1H), 7.62–7.55 (m, 1H), 7.49–7.40 (m, 3H), 7.36 (d, *J* = 7.9 Hz, 1H), 7.32 (d, *J* = 7.6 Hz, 1H), 7.22–7.10 (m, 3H), 6.69 (d, *J* = 2.5 Hz, 1H), 6.17–6.12 (m, 1H), 4.72 (s, 1H), 4.11–4.01 (m, 1H), 3.86–3.81 (m, 1H), 3.67–3.61 (m, 1H), 3.41–3.35 (m, 1H), 3.22–3.10 (m, 1H), 2.57–2.46 (m, 1H), 2.28–2.16 (m, 2H), 1.52 (s, 3H). ^13^C-NMR (150 MHz, CDCl_3_) *δ* 180.6, 176.6, 171.9, 141.2, 139.7, 135.6, 133.8, 133.2, 132.7, 132.6, 132.2, 131.4, 131.1, 130.1, 130.0, 129.8, 129.1, 129.1, 128.9, 127.4, 126.7, 126.5, 125.9, 125.1, 124.7, 122.7, 74.3, 73.8, 57.1, 55.1, 41.6, 21.0, 18.1. LRMS (ESI, *m*/*z*): 765.9 [M − H]^−^. HRMS (ESI, *m*/*z*): calcd for C_34_H_26_BrCl_3_N_3_NiO_3_, 765.9582 [M − H]^−^; found 765.9579. The dr was determined by LC-MS using an Eclipse XDB-C18 column (250 × 4.6 mm, 5 µm) (CH_3_CN/ H_2_O = 65:35, flow rate 1.0 mL/min, λ = 254 nm), t_major_ = 25.652 min, t_minor_ = 28.939, dr = 98:2.

*Nickel(II)-(S)-N-(2-benzoyl-4-chlorophenyl)-1-(3,4-dichlorobenzyl)-2-methylpyrrolidine-2-carboxamide/(S)-2-cyclobutylglycine Schiff Base Complex* (**6l**). Red solid (118 mg, yield 89%); mp 307–308 °C; [*α*]^20^_D_ = +3252 (c = 1, CHCl_3_); ^1^H-NMR (400 MHz, CDCl_3_) *δ* 9.12 (s, 1H), 8.23 (d, *J* = 9.3 Hz, 1H), 7.70 (d, *J* = 8.2 Hz, 1H), 7.60–7.38 (m, 3H), 7.29 (t, *J* = 6.9 Hz, 2H), 7.09 (dd, *J* = 9.3, 2.5 Hz, 1H), 6.87 (d, *J* = 7.4 Hz, 1H), 6.58 (d, *J* = 2.5 Hz, 1H), 4.21–3.95 (m, 1H), 3.82–3.69 (m, 2H), 3.51 (d, *J* = 13.0 Hz, 1H), 3.40 (t, *J* = 8.7 Hz, 1H), 3.17–2.88 (m, 1H), 2.67 (q, *J* = 17.3 Hz, 1H), 2.58–2.33 (m, 3H), 2.29–2.12 (m, 2H), 2.09–1.96 (m, 1H), 1.81 (dd, *J* = 18.1, 8.2 Hz, 1H), 1.70 (dd, *J* = 18.3, 9.0 Hz, 2H), 1.44 (s, 3H). ^13^C-NMR (125 MHz, CDCl_3_) *δ* 180.3, 177.2, 169.6, 140.9, 135.8, 133.9, 133.3, 133.1, 133.0, 132.5, 132.3, 131.1, 130.3, 129.8, 129.4, 129.1, 128.1, 128.0, 127.8, 125.8, 124.2, 74.5, 74.0, 57.2, 55.0, 41.9, 40.7, 26.0, 25.6, 20.7, 18.1, 17.5. LRMS (ESI, *m*/*z*): 668.0 [M + H]^+^. HRMS (ESI, *m*/*z*): calcd for C_32_H_31_Cl_3_N_3_NiO_3_, 668.0604 [M + H]^+^; found 668.0624. The dr was determined by LC-MS using an Eclipse XDB-C18 column (250 × 4.6 mm, 5 µm) (CH_3_CN/H_2_O = 65:35, flow rate 1.0 mL/min, λ = 254 nm), t_major_ = 16.283 min, t_minor_ = not found, dr > 99:1.

*Nickel(II)-(S)-N-(2-benzoyl-4-chlorophenyl)-1-(3,4-dichlorobenzyl)-2-methylpyrrolidine-2-carboxamide/(S)-2-amino-4,4,4-trifluorobutyric acid Schiff Base Complex* (**6m**). Red solid (121 mg, yield 87%); mp 244–246 °C; [*α*]^20^_D_ = +2830 (c = 1, CHCl_3_); ^1^H-NMR (400 MHz, CDCl_3_) *δ* 9.17 (d, *J* = 1.8 Hz, 1H), 8.35 (d, *J* = 9.3 Hz, 1H), 7.66–7.55 (m, 3H), 7.55–7.50 (m, 1H), 7.36 (d, *J* = 8.1 Hz, 1H), 7.30 (d, *J* = 6.8 Hz, 1H), 7.17 (dd, *J* = 9.3, 2.5 Hz, 1H), 6.90 (d, *J* = 7.4 Hz, 1H), 6.61 (d, *J* = 2.5 Hz, 1H), 4.23–4.11 (m, 2H), 3.74–3.57 (m, 2H), 3.39 (t, *J* = 8.7 Hz, 1H), 3.06–2.92 (m, 1H), 2.63–2.37 (m, 2H), 2.21–2.10 (m, 2H), 2.05–1.93 (m, 1H), 1.44 (s, 3H). ^13^C-NMR (125 MHz, CDCl_3_) *δ* 180.4, 177.2, 172.4, 141.9, 135.8, 134.0, 133.2, 133.2, 133.2, 133.1, 132.5, 131.3, 130.8, 129.9, 129.8, 129.7, 127.6, 127.6, 126.8, 125.8, 124.6, 74.9, 64.5, 57.8, 54.8, 41.2, 35.6 (q, *J* = 28.9 Hz), 20.3, 17.4. LRMS (ESI, *m*/*z*): 694.0 [M − H]^−^. HRMS (ESI, *m*/*z*): calcd for C_30_H_24_Cl_3_F_3_N_3_NiO_3_, 694.0194 [M − H]^−^; found 694.0203. The dr was determined by LC-MS using an Eclipse XDB-C18 column (250 × 4.6 mm, 5 µm) (CH_3_CN/H_2_O = 65:35, flow rate 1.0 mL/min, λ = 254 nm), t_major_ = 13.938 min, t_minor_ = 15.440, dr = 99:1.

*Nickel(II)-(S)-N-(2-benzoyl-4-chlorophenyl)-1-(3,4-dichlorobenzyl)-2-methylpyrrolidine-2-carboxamide/(S)-norvaline Schiff Base Complex* (**6n**). Red solid (121 mg, yield 92%); mp 244–247 °C; [*α*]^20^_D_ = +2842 (c = 1, CHCl_3_); ^1^H-NMR (400 MHz, CDCl_3_) *δ* 9.14–9.12 (m, 1H), 8.22 (d, *J* = 9.3 Hz, 1H), 7.75 (dd, *J* = 8.2, 2.0 Hz, 1H), 7.60–7.51 (m, 2H), 7.50–7.45 (m, 1H), 7.35 (d, *J* = 8.2 Hz, 1H), 7.33–7.29 (m, 1H), 7.13 (dd, *J* = 9.3, 2.6 Hz, 1H), 6.89–6.85 (m, 1H), 6.62 (d, *J* = 2.6 Hz, 1H), 4.09–3.99 (m, 1H), 3.89 (dd, *J* = 7.6, 3.6 Hz, 1H), 3.77 (d, *J* = 13.0 Hz, 1H), 3.57 (d, *J* = 13.1 Hz, 1H), 3.40 (t, *J* = 8.8 Hz, 1H), 3.22–3.10 (m, 1H), 2.50–2.38 (m, 1H), 2.29–2.16 (m, 2H), 2.15–1.99 (m, 1H), 1.95–1.82 (m, 1H), 1.71 (dd, *J* = 13.1, 6.3 Hz, 1H), 1.62–1.56 (m, 1H), 1.47 (s, 3H), 0.81 (t, *J* = 7.2 Hz, 3H). ^13^C-NMR (150 MHz, CDCl_3_) *δ* 180.2, 178.8, 169.6, 140.7, 135.6, 133.7, 133.0, 132.9, 132.8, 132.1, 131.0, 130.1, 129.7, 129.3, 129.1, 127.9, 127.3, 127.3, 125.7, 124.3, 74.3, 70.3, 57.0, 54.8, 41.4, 37.2, 20.8, 18.2, 17.7, 13.7. LRMS (ESI, *m*/*z*): 654.1 [M − H]^−^. HRMS (ESI, *m*/*z*): calcd for C_31_H_29_Cl_3_N_3_NiO_3_, 654.0633 [M − H]^−^; found 654.0643. The dr was determined by LC-MS using an Eclipse XDB-C18 column (250 × 4.6 mm, 5 µm) (CH_3_CN/H_2_O = 65:35, flow rate 1.0 mL/min, λ = 254 nm), t_major_ = 14.871 min, t_minor_ = not found, dr > 99:1.

*Nickel(II)-(S)-N-(2-benzoyl-4-chlorophenyl)-1-(3,4-dichlorobenzyl)-2-methylpyrrolidine-2-carboxamide/(S)-valine Schiff Base Complex* (**6o**). Red solid (122 mg, yield 93%); mp 267–269 °C; [*α*]^20^_D_ = +3016 (c = 1, CHCl_3_); ^1^H-NMR (400 MHz, CDCl_3_) *δ* 9.17 (d, *J* = 2.1 Hz, 1H), 8.28 (d, *J* = 9.3 Hz, 1H), 7.65 (dd, *J* = 8.2, 2.1 Hz, 1H), 7.54 (dt, *J* = 16.1, 7.9 Hz, 2H), 7.45 (t, *J* = 6.9 Hz, 1H), 7.31 (dd, *J* = 7.7, 4.2 Hz, 2H), 7.10 (dd, *J* = 9.3, 2.6 Hz, 1H), 6.85 (d, *J* = 7.5 Hz, 1H), 6.61 (d, *J* = 2.6 Hz, 1H), 4.14–3.97 (m, 1H), 3.83–3.72 (m, 2H), 3.53 (d, *J* = 13.1 Hz, 1H), 3.40 (t, *J* = 9.0 Hz, 1H), 3.00 (td, *J* = 12.5, 4.0 Hz, 1H), 2.53–2.34 (m, 1H), 2.27–2.08 (m, 2H), 1.84 (d, *J* = 6.6 Hz, 3H), 1.80–1.67 (m, 1H), 1.45 (s, 3H), 0.73 (d, *J* = 6.8 Hz, 3H). ^13^C-NMR (125 MHz, CDCl_3_) *δ* 180.4, 177.4, 170.3, 141.0, 135.9, 134.0, 133.2, 133.2, 133.0, 132.5, 132.4, 131.2, 130.2, 129.8, 129.5, 129.1, 127.9, 127.9, 127.5, 125.8, 124.2, 75.6, 74.4, 57.3, 54.9, 41.7, 34.4, 20.5, 19.9, 17.8, 17.7. LRMS (ESI, *m*/*z*): 656.0 [M + H]^+^. HRMS (ESI, *m*/*z*): calcd for C_31_H_31_Cl_3_N_3_NiO_3_, 656.0604 [M + H]^+^; found 656.0606. The dr was determined by LC-MS with binary pump, photodiode array detector (DAD), using Eclipse XDB-C18 column (250 × 4.6 mm, 5 µm) (CH_3_CN/H_2_O = 65:35, flow rate 1.0 mL/min, λ = 254 nm), t_major_ = 14.594 min, t_minor_ = not found, dr > 99:1.

*Nickel(II)-(S)-N-(2-benzoyl-4-chlorophenyl)-1-(3,4-dichlorobenzyl)-2-methylpyrrolidine-2-carboxamide/(S)-leucine Schiff Base Complex* (**6p**). Red solid (127 mg, yield 95%); mp 156–159 °C; [*α*]^20^_D_ = +2731 (c = 1, CHCl_3_); ^1^H-NMR (500 MHz, CDCl_3_) *δ* 9.14–9.11 (m, 1H), 8.16 (d, *J* = 9.3 Hz, 1H), 7.80 (dd, *J* = 8.2, 2.1 Hz, 1H), 7.59–7.55 (m, 1H), 7.53 (t, *J* = 7.4 Hz, 1H), 7.50–7.45 (m, 1H), 7.36–7.32 (m, 2H), 7.12 (dd, *J* = 9.3, 2.6 Hz, 1H), 6.88 (d, *J* = 7.6 Hz, 1H), 6.62 (d, *J* = 2.6 Hz, 1H), 4.06–3.97 (m, 1H), 3.85 (dd, *J* = 10.5, 4.1 Hz, 1H), 3.82 (d, *J* = 13.2 Hz, 1H), 3.54 (d, *J* = 13.1 Hz, 1H), 3.40 (t, *J* = 8.8 Hz, 1H), 3.34–3.17 (m, 1H), 2.48–2.34 (m, 2H), 2.28–2.18 (m, 2H), 1.97–1.86 (m, 1H), 1.47 (s, 3H), 1.39–1.32 (m, 1H), 0.83 (d, *J* = 6.7 Hz, 3H), 0.40 (d, *J* = 6.5 Hz, 3H). ^13^C-NMR (125 MHz, CDCl_3_) *δ* 180.2, 178.6, 169.0, 140.6, 135.6, 133.7, 133.1, 132.9, 132.7, 132.0, 130.9, 130.1, 129.7, 129.2, 129.1, 127.9, 127.6, 127.4, 125.7, 124.3, 74.1, 69.1, 56.7, 55.1, 46.0, 41.6, 24.3, 23.7, 21.2, 20.8, 17.9. LRMS (ESI, *m*/*z*): 668.0 [M − H]^−^. HRMS (ESI, *m*/*z*): calcd for C_32_H_31_Cl_3_N_3_NiO_3_, 668.0955 [M − H]^−^; found 668.0947. The dr was determined by LC-MS using an Eclipse XDB-C18 column (250 × 4.6 mm, 5 µm) (CH_3_CN/H_2_O = 65:35, flow rate 1.0 mL/min, λ = 254 nm), t_major_ = 20.065 min, t_minor_ = not found, dr > 99:1.

*Nickel(II)-(S)-N-(2-benzoyl-4-chlorophenyl)-1-(3,4-dichlorobenzyl)-2-methylpyrrolidine-2-carboxamide/(S)-methionine Schiff Base Complex* (**6q**). Red solid (127 mg, yield 93%); mp 143–145 °C; [*α*]^20^_D_ = +1883 (c = 1, CHCl_3_); ^1^H-NMR (400 MHz, CDCl_3_) *δ* 9.15–9.11 (m, 1H), 8.23 (d, *J* = 9.3 Hz, 1H), 7.76–7.71 (m, 1H), 7.62–7.52 (m, 2H), 7.52–7.46 (m, 1H), 7.38–7.34 (m, 1H), 7.34–7.30 (m, 1H), 7.14 (dd, *J* = 9.3, 2.5 Hz, 1H), 6.90 (d, *J* = 7.5 Hz, 1H), 6.62 (d, *J* = 2.5 Hz, 1H), 4.07–3.98 (m, 1H), 3.97–3.92 (m, 1H), 3.81–3.72 (m, 1H), 3.61–3.54 (m, 1H), 3.39 (t, *J* = 8.9 Hz, 1H), 3.23–3.12 (m, 1H), 3.11–3.03 (m, 1H), 2.65–2.54 (m, 1H), 2.50–2.38 (m, 1H), 2.30–2.14 (m, 3H), 1.99 (s, 3H), 1.93–1.83 (m, 1H), 1.47 (s, 3H). ^13^C-NMR (150 MHz, CDCl_3_) *δ* 180.4, 178.3, 170.3, 141.1, 135.7, 133.9, 133.3, 133.1, 132.9, 132.5, 132.4, 131.2, 130.4, 129.9, 129.6, 129.6, 129.4, 127.9, 127.5, 127.4, 125.9, 124.6, 74.5, 69.8, 57.3, 55.0, 41.6, 35.1, 29.8, 21.2, 18.0, 15.8. LRMS (ESI, *m*/*z*): 686.0 [M − H]^−^. HRMS (ESI, *m*/*z*): calcd for C_31_H_29_Cl_3_N_3_NiO_3_S, 686.0354 [M − H]^−^; found 686.0371. The dr was determined by LC-MS using an Eclipse XDB-C18 column (250 × 4.6 mm, 5 µm) (CH_3_CN/H_2_O = 65:35, flow rate 1.0 mL/min, λ = 254 nm), t_major_ = 13.877 min, t_minor_ = 15.012, dr = 99:1.

*Nickel(II)-(R)-N-(2-benzoyl-4-chlorophenyl)-1-(3,4-dichlorobenzyl)-2-methylpyrrolidine-2-carboxamide/(R)-phenylalanine Schiff Base Complex* (**6a**). Red solid (128 mg, yield 91%); mp 121–124 °C; [*α*]^20^_D_ = –1684 (c = 1, CHCl_3_); ^1^H-NMR (400 MHz, CDCl_3_) *δ* 9.09 (d, *J* = 2.1 Hz, 1H), 8.34 (d, *J* = 9.3 Hz, 1H), 7.62–7.56 (m, 2H), 7.56–7.51 (m, 1H), 7.43–7.32 (m, 5H), 7.31–7.29 (m, 1H), 7.16–7.09 (m, 3H), 6.68–6.64 (m, 1H), 6.62 (d, *J* = 2.6 Hz, 1H), 4.24 (t, *J* = 5.2 Hz, 1H), 3.65–3.60 (m, 1H), 3.51–3.45 (m, 1H), 3.36–3.26 (m, 1H), 3.20–3.09 (m, 2H), 2.83–2.76 (m, 1H), 2.41–2.27 (m, 1H), 2.16–2.05 (m, 1H), 2.05–1.95 (m, 1H), 1.88–1.77 (m, 1H), 1.35 (s, 3H). ^13^C-NMR (125 MHz, CDCl_3_) *δ* 180.3, 177.9, 170.6, 141.3, 135.6, 133.8, 133.2, 132.9, 132.8, 132.4, 132.4, 130.9, 130.3, 130.1, 129.6, 129.3, 129.1, 128.9, 127.7, 127.5, 127.3, 125.6, 124.1, 74.1, 71.6, 57.1, 54.5, 41.2, 39.6, 20.5, 17.6. LRMS (ESI, *m*/*z*): 702.0 [M − H]^−^. HRMS (ESI, *m*/*z*): calcd for C_35_H_29_Cl_3_N_3_NiO_3_, 702.0633 [M − H]^−^; found 702.0650. The dr was determined by LC-MS using an Eclipse XDB-C18 column (250 × 4.6 mm, 5 µm) (CH_3_CN/H_2_O = 65:35, flow rate 1.0 mL/min, λ = 254 nm), t_major_ = 15.665 min, t_minor_ = not found, dr > 99:1.

*Nickel(II)-(R)-N-(2-benzoyl-4-chlorophenyl)-1-(3,4-dichlorobenzyl)-2-methylpyrrolidine-2-carboxamide/(R)-2-methoxyphenylalanine Schiff Base Complex* (**6b**). Red solid (129 mg, yield 88%); mp 112–114 °C; [*α*]^20^_D_ = –1818 (c = 1, CHCl_3_); ^1^H-NMR (500 MHz, CDCl_3_) *δ* 9.01 (d, *J* = 2.0 Hz, 1H), 8.37 (d, *J* = 9.3 Hz, 1H), 7.60–7.53 (m, 2H), 7.51 (t, *J* = 7.5 Hz, 1H), 7.39 (t, *J* = 7.4 Hz, 1H), 7.36–7.32 (m, 1H), 7.30 (d, *J* = 8.1 Hz, 2H), 7.21 (dd, *J* = 7.4, 1.3 Hz, 1H), 7.11 (dd, *J* = 9.3, 2.6 Hz, 1H), 6.97 (t, *J* = 7.4 Hz, 1H), 6.86 (dd, *J* = 11.1, 8.1 Hz, 2H), 6.63 (d, *J* = 2.6 Hz, 1H), 4.15 (t, *J* = 4.9 Hz, 1H), 3.52 (dd, *J* = 30.2, 13.0 Hz, 2H), 3.31–3.23 (m, 4H), 3.19 (dd, *J* = 13.6, 5.0 Hz, 1H), 3.09 (t, *J* = 8.9 Hz, 1H), 2.91 (dd, *J* = 13.6, 4.9 Hz, 1H), 2.33–2.20 (m, 1H), 2.10–2.02 (m, 1H), 1.95 (dd, *J* = 19.5, 9.9 Hz, 1H), 1.84–1.72 (m, 1H), 1.32 (s, 3H). ^13^C-NMR (125 MHz, CDCl_3_) *δ* 180.4, 178.4, 170.9, 158.3, 141.2, 135.8, 134.0, 133.5, 133.0, 132.9, 132.7, 132.6, 132.3, 131.0, 130.0, 129.8, 129.3, 129.2, 129.2, 128.3, 127.9, 127.6, 125.6, 124.3, 124.1, 121.3, 110.5, 74.5, 71.9, 57.5, 54.7, 54.5, 41.3, 34.6, 20.8, 17.9. LRMS (ESI, *m*/*z*): 734.0 [M + H]^+^. HRMS (ESI, *m*/*z*): calcd for C_36_H_33_Cl_3_N_3_NiO_4_, 734.0885, [M + H]^+^; found 734.0887. The dr was determined by LC-MS using an Eclipse XDB-C18 column (250 × 4.6 mm, 5 µm) (CH_3_CN/H_2_O = 65:35, flow rate 1.0 mL/min, λ = 254 nm), t_major_ = 19.227 min, t_minor_ = not found, dr > 99:1.

*Nickel(II)-(R)-N-(2-benzoyl-4-chlorophenyl)-1-(3,4-dichlorobenzyl)-2-methylpyrrolidine-2-carboxamide/(R)-3-methoxyphenylalanine Schiff Base Complex* (**6c**). Red solid (131 mg, yield 93%); mp 104–106 °C; [*α*]^20^_D_ = –2923 (c = 1, CHCl_3_); ^1^H-NMR (400 MHz, CDCl_3_) *δ* 9.08 (d, *J* = 2.0 Hz, 1H), 8.33 (d, *J* = 9.3 Hz, 1H), 7.62–7.47 (m, 3H), 7.40–7.28 (m, 3H), 7.26–7.18 (m, 1H), 7.12 (dd, *J* = 9.3, 2.6 Hz, 1H), 6.88 (dd, *J* = 8.3, 2.3 Hz, 1H), 6.67 (d, *J* = 7.5 Hz, 1H), 6.60 (d, *J* = 2.6 Hz, 2H), 6.55 (d, *J* = 7.9 Hz, 1H), 4.21 (t, *J* = 5.3 Hz, 1H), 3.65 (s, 3H), 3.55 (dd, *J* = 59.3, 13.1 Hz, 2H), 3.45–3.31 (m, 1H), 3.26–3.03 (m, 2H), 2.81 (dd, *J* = 13.8, 5.1 Hz, 1H), 2.46 (dq, *J* = 14.1, 7.2 Hz, 1H), 2.25–2.11 (m, 1H), 2.07–1.96 (m, 1H), 1.95–1.78 (m, 1H), 1.35 (s, 3H). ^13^C-NMR (125 MHz, CDCl_3_) *δ* 180.6, 178.3, 170.9, 141.5, 135.8, 135.7, 134.0, 133.4, 133.1, 133.0, 132.7, 132.6, 131.1, 130.3, 129.8, 129.6, 129.5, 129.3, 127.7, 127.4, 126.4, 125.8, 124.3, 124.3, 74.4, 71.3, 57.4, 54.8, 41.5, 34.1, 20.9, 17.9. LRMS (ESI, *m*/*z*): 734.0 [M + H]^+^. HRMS (ESI, *m*/*z*): calcd for C_36_H_33_Cl_3_N_3_NiO_4_, 734.0711 [M + H]^+^; found 734.0718. The dr was determined by LC-MS using an Eclipse XDB-C18 column (250 × 4.6 mm, 5 µm) (CH_3_CN/H_2_O = 65:35, flow rate 1.0 mL/min, λ = 254 nm), t_major_ = 16.492 min, t_minor_ = not found, dr > 99:1.

*Nickel(II)-(R)-N-(2-benzoyl-4-chlorophenyl)-1-(3,4-dichlorobenzyl)-2-methylpyrrolidine-2-carboxamide/(R)-3-methylphenylalanine Schiff Base Complex* (**6d**). Red solid (129 mg, yield 90%); mp 105–107 °C; [*α*]^20^_D_ = –1898 (c = 1, CHCl_3_); ^1^H-NMR (400 MHz, CDCl_3_) *δ* 9.06 (d, *J* = 1.5 Hz, 1H), 8.34 (d, *J* = 9.3 Hz, 1H), 7.55 (dt, *J* = 17.5, 7.4 Hz, 3H), 7.38 (t, *J* = 7.5 Hz, 1H), 7.31 (t, *J* = 6.7 Hz, 2H), 7.23 (t, *J* = 7.5 Hz, 1H), 7.17–7.09 (m, 2H), 6.87 (d, *J* = 7.8 Hz, 2H), 6.63 (dd, *J* = 12.2, 5.1 Hz, 2H), 4.21 (t, *J* = 5.1 Hz, 1H), 3.53 (dd, *J* = 47.7, 13.1 Hz, 2H), 3.31 (td, *J* = 12.6, 5.8 Hz, 1H), 3.11 (dd, *J* = 13.7, 4.7 Hz, 2H), 2.75 (dd, *J* = 13.7, 5.4 Hz, 1H), 2.47–2.31 (m, 1H), 2.27 (s, 3H), 2.13–2.06 (m, 1H), 1.99 (dd, *J* = 19.4, 9.7 Hz, 1H), 1.84 (dt, *J* = 14.9, 6.7 Hz, 1H), 1.33 (s, 3H). ^13^C-NMR (125 MHz, CDCl_3_) *δ* 180.4, 178.2, 170.7, 141.5, 138.7, 135.8, 135.6, 134.0, 133.3, 133.1, 133.0, 132.6, 132.6, 131.3, 131.1, 130.2, 129.8, 129.5, 129.2, 128.9, 128.4, 127.9, 127.7, 127.5, 127.5, 125.7, 124.2, 74.3, 71.8, 57.3, 54.7, 41.2, 39.8, 21.6, 20.6, 17.8. LRMS (ESI, *m*/*z*): 718.0 [M + H]^+^. HRMS (ESI, *m*/*z*): calcd for C_36_H_33_Cl_3_N_3_NiO_3_, 718.0762 [M + H]^+^; found 718.0773. The dr was determined by LC-MS using an Eclipse XDB-C18 column (250 × 4.6 mm, 5 µm) (CH_3_CN/H_2_O = 65:35, flow rate 1.0 mL/min, λ = 254 nm), t_major_ = 21.866 min, t_minor_ = not found, dr > 99:1.

*Nickel(II)-(R)-N-(2-benzoyl-4-chlorophenyl)-1-(3,4-dichlorobenzyl)-2-methylpyrrolidine-2-carboxamide/(R)-4-fluorophenylalanine Schiff Base Complex* (**6e**). Red solid (129 mg, yield 89%); mp 235–237 °C; [*α*]^20^_D_ = −2300 (c = 1, CHCl_3_); ^1^H-NMR (400 MHz, CDCl_3_) *δ* 9.06 (d, *J* = 1.5 Hz, 1H), 8.32 (d, *J* = 9.3 Hz, 1H), 7.56 (p, *J* = 7.5 Hz, 3H), 7.43 (t, *J* = 7.3 Hz, 1H), 7.37–7.28 (m, 2H), 7.13 (dd, *J* = 9.3, 2.5 Hz, 1H), 7.05 (d, *J* = 7.0 Hz, 4H), 6.73 (d, *J* = 7.5 Hz, 1H), 6.62 (d, *J* = 2.5 Hz, 1H), 4.21 (t, *J* = 5.0 Hz, 1H), 3.54 (dd, *J* = 47.7, 13.0 Hz, 2H), 3.34 (td, *J* = 12.6, 6.0 Hz, 1H), 3.20–3.01 (m, 2H), 2.74 (dd, *J* = 14.0, 5.4 Hz, 1H), 2.45–2.26 (m, 1H), 2.26–2.12 (m, 1H), 2.00 (dd, *J* = 19.3, 9.6 Hz, 1H), 1.90–1.83 (m, 1H), 1.35 (s, 3H). ^13^C-NMR (125 MHz, CDCl_3_) *δ* 180.5, 177.9, 170.9, 163.4, 161.8, 141.5, 135.7, 133.9, 133.3, 133.1, 133.0, 132.7, 132.6, 131.9, 131.9, 131.5, 131.5, 131.1, 130.4, 129.8, 129.6, 129.3, 127.7, 127.6, 127.4, 125.9, 124.3, 116.0, 115.8, 74.3, 71.6, 57.4, 54.6, 41.2, 38.8, 20.7, 17.8. LRMS (ESI, *m*/*z*): 722.0 [M + H]^+^. HRMS (ESI, *m*/*z*): calcd for C_35_H_30_Cl_3_FN_3_NiO_3_, 722.0511 [M + H]^+^; found 722.0517. The dr was determined by LC-MS using an Eclipse XDB-C18 column (250 × 4.6 mm, 5 µm) (CH_3_CN/H_2_O = 65:35, flow rate 1.0 mL/min, λ = 254 nm), t_major_ = 18.447 min, t_minor_ = not found, dr > 99:1.

*Nickel(II)-(R)-N-(2-benzoyl-4-chlorophenyl)-1-(3,4-dichlorobenzyl)-2-methylpyrrolidine-2-carboxamide/(R)-3,5-diiodotyrosine Schiff Base Complex* (**6f**): red solid (169 mg, yield 87%); mp 233–235 °C; [*α*]^20^_D_ = –1774 (c = 1, CHCl_3_); ^1^H-NMR (400 MHz, CDCl_3_) *δ* 9.08–9.05 (m, 1H), 8.47–8.42 (m, 1H), 7.63–7.59 (m, 2H), 7.59–7.55 (m, 1H), 7.52–7.46 (m, 1H), 7.37–7.32 (m, 2H), 7.29–7.27 (m, 2H), 7.20–7.15 (m, 1H), 6.79 (d, *J* = 7.4 Hz, 1H), 6.67 (d, *J* = 2.5 Hz, 1H), 5.83 (s, 1H), 4.14–4.10 (m, 1H), 3.57 (q, *J* = 13.8 Hz,2H), 3.48–3.38 (m, 1H), 3.23–3.16 (m, 1H), 2.94–2.87 (m, 1H), 2.73–2.65 (m, 1H), 2.56–2.46 (m, 1H), 2.31–2.19 (m, 1H), 2.10–1.96 (m, 2H), 1.38 (s, 3H). ^13^C-NMR (150 MHz, CDCl_3_) *δ* 180.4, 177.6, 171.0, 153.4, 141.7, 140.6, 135.6, 134.0, 133.2, 133.1, 133.0, 132.6, 131.8, 131.1, 130.6, 129.8, 129.7, 129.4, 127.6, 127.5, 127.4, 125.8, 124.4, 82.8, 74.5, 71.2, 57.5, 54.6, 41.0, 38.0, 21.0, 17.9. LRMS (ESI, *m*/*z*): 969.8 [M − H]^−^. HRMS (ESI, *m*/*z*): calcd for C_35_H_28_Cl_3_I_2_N_3_NiO_4_, 969.8515 [M − H]^−^; found 969.8511. The dr was determined by LC-MS using an Eclipse XDB-C18 column (250 × 4.6 mm, 5 µm) (CH_3_CN/H_2_O = 65:35, flow rate 1.0 mL/min, λ = 254 nm), t_major_ = 17.454 min, t_minor_ = not found, dr > 99:1.

*Nickel(II)-(R)-N-(2-benzoyl-4-chlorophenyl)-1-(3,4-dichlorobenzyl)-2-methylpyrrolidine-2-carboxamide/(R)-3-(1-naphthyl)alanine Schiff Base Complex* (**6g**). Red solid (142 mg, yield 94%); mp 114–116 °C; [*α*]^20^_D_ = −1812 (c = 1, CHCl_3_); ^1^H-NMR (400 MHz, CDCl_3_) *δ* 9.09 (d, *J* = 2.0 Hz, 1H), 8.30 (d, *J* = 9.3 Hz, 1H), 7.78 (t, *J* = 8.0 Hz, 2H), 7.68 (dd, *J* = 8.2, 2.0 Hz, 1H), 7.57 (d, *J* = 8.6 Hz, 1H), 7.43–7.35 (m, 1H), 7.30 (dd, *J* = 15.6, 7.6 Hz, 4H), 7.23–7.11 (m, 3H), 7.08 (dd, *J* = 9.3, 2.6 Hz, 1H), 6.68 (t, *J* = 7.6 Hz, 1H), 6.44 (d, *J* = 2.6 Hz, 1H), 5.64 (d, *J* = 7.7 Hz, 1H), 4.41 (dd, *J* = 8.3, 4.3 Hz, 1H), 4.07 (dd, *J* = 14.2, 8.5 Hz, 1H), 3.79 (dd, *J* = 14.1, 4.3 Hz, 1H), 3.70 (d, *J* = 13.2 Hz, 1H), 3.61 (td, *J* = 13.1, 5.9 Hz, 1H), 3.46 (d, *J* = 13.2 Hz, 1H), 3.24 (t, *J* = 9.1 Hz, 1H), 2.83–2.53 (m, 1H), 2.39–2.22 (m, 1H), 2.12 (dd, *J* = 19.8, 9.6 Hz, 1H), 2.00 (dt, *J* = 15.0, 8.9 Hz, 1H), 1.40 (s, 3H). ^13^C-NMR (125 MHz, CDCl_3_) *δ* 180.3, 178.3, 170.5, 141.3, 135.8, 134.2, 133.8, 133.3, 133.0, 132.8, 132.6, 132.5, 132.3, 131.7, 131.2, 129.8, 129.6, 128.8, 128.7, 128.6, 128.6, 128.5, 127.7, 127.5, 127.4, 126.5, 126.1, 125.7, 125.6, 124.1, 123.4, 74.4, 71.6, 56.9, 55.1, 41.6, 40.2, 20.9, 17.9. LRMS (ESI, *m*/*z*): 754.0 [M + H]^+^. HRMS (ESI, *m*/*z*): calcd for C_39_H_33_Cl_3_N_3_NiO_3_, 754.0763 [M + H]^+^; found 754.0779. The dr was determined by LC-MS using an Eclipse XDB-C18 column (250 × 4.6 mm, 5 µm) (CH_3_CN/H_2_O = 65:35, flow rate 1.0 mL/min, λ = 254 nm), t_major_ = 23.723 min, t_minor_ = not found, dr > 99:1.

*Nickel(II)-(R)-N-(2-benzoyl-4-chlorophenyl)-1-(3,4-dichlorobenzyl)-2-methylpyrrolidine-2-carboxamide/(R)-3-(3-benzothienyl)alanine Schiff Base Complex* (**6h**). Red solid (148 mg, yield 98%); mp 127–129 °C; [*α*]^20^_D_ = –1836 (c = 1, CHCl_3_); ^1^H-NMR (400 MHz, CDCl_3_) *δ* 9.06 (d, *J* = 2.0 Hz, 1H), 8.34 (d, *J* = 9.3 Hz, 1H), 7.86 (d, *J* = 8.0 Hz, 1H), 7.57 (dd, *J* = 8.2, 2.0 Hz, 1H), 7.49 (t, *J* = 7.5 Hz, 1H), 7.45–7.34 (m, 2H), 7.30 (dd, *J* = 11.9, 6.2 Hz, 3H), 7.19–7.08 (m, 4H), 6.59 (d, *J* = 2.6 Hz, 1H), 6.48 (d, *J* = 7.5 Hz, 1H), 4.31 (t, *J* = 5.5 Hz, 1H), 3.71–3.59 (m, 1H), 3.51 (dd, *J* = 57.6, 9.8 Hz, 2H), 3.18 (dd, *J* = 14.5, 4.7 Hz, 1H), 3.11–2.97 (m, 2H), 2.12–1.90 (m, 3H), 1.81–1.69 (m, 1H), 1.30 (s, 3H). ^13^C-NMR (125 MHz, CDCl_3_) *δ* 180.3, 178.3, 170.8, 141.5, 140.6, 139.3, 135.7, 133.9, 133.2, 133.0, 132.9, 132.7, 131.1, 130.5, 130.1, 129.8, 129.3, 129.0, 127.5, 127.5, 125.9, 125.8, 124.8, 124.5, 124.3, 122.9, 122.1, 74.3, 70.9, 57.2, 54.8, 41.0, 34.0, 20.4, 17.8. LRMS (ESI, *m*/*z*): 760.0 [M + H]^+^. HRMS (ESI, *m*/*z*): calcd for C_37_H_31_Cl_3_N_3_NiO_3_S, 760.0326 [M + H]^+^; found 760.0333. The dr was determined by LC-MS using an Eclipse XDB-C18 column (250 × 4.6 mm, 5 µm) (CH_3_CN/H_2_O = 65:35, flow rate 1.0 mL/min, λ = 254 nm), t_major_ = 27.103 min, t_minor_ = not dound, dr > 99:1.

*Nickel(II)-(R)-N-(2-benzoyl-4-chlorophenyl)-1-(3,4-dichlorobenzyl)-2-methylpyrrolidine-2-carboxamide/(R)-3-(3-thienyl)alanine Schiff Base Complex* (**6i**). Red solid (128 mg, yield 92%); mp 208–210 °C; [*α*]^20^_D_ = −2419 (c = 1, CHCl_3_); ^1^H-NMR (400 MHz, CDCl_3_) *δ* 9.09 (d, *J* = 2.1 Hz, 1H), 8.34 (d, *J* = 9.3 Hz, 1H), 7.63–7.49 (m, 3H), 7.46–7.39 (m, 1H), 7.38–7.28 (m, 3H), 7.13 (dd, *J* = 9.3, 2.6 Hz, 1H), 7.04 (d, *J* = 1.8 Hz, 1H), 6.86 (dd, *J* = 4.9, 1.2 Hz, 1H), 6.70 (d, *J* = 7.7 Hz, 1H), 6.62 (d, *J* = 2.6 Hz, 1H), 4.18 (t, *J* = 5.1 Hz, 1H), 3.63 (d, *J* = 13.1 Hz, 1H), 3.50 (dq, *J* = 11.9, 5.9 Hz, 2H), 3.25–3.05 (m, 2H), 2.77 (dd, *J* = 14.3, 5.4 Hz, 1H), 2.62–2.41 (m, 1H), 2.29–2.12 (m, 1H), 2.09–1.98 (m, 1H), 1.96–1.84 (m, 1H), 1.36 (s, 3H). ^13^C-NMR (125 MHz, CDCl_3_) *δ* 180.6, 178.3, 170.9, 141.5, 135.8, 135.7, 134.0, 133.4, 133.1, 133.0, 132.7, 132.6, 131.1, 130.3, 129.8, 129.6, 129.5, 129.3, 127.7, 127.4, 126.4, 125.8, 124.3, 124.3, 74.4, 71.3, 57.4, 54.8, 41.5, 34.1, 20.9, 17.9. LRMS (ESI, *m/z*): 710.0 [M + H]^+^. HRMS (ESI, *m/z*): calcd for C_33_H_29_Cl_3_N_3_NiO_3_S, 710.0168 [M + H]^+^; found 710.0175. The dr was determined by LC-MS using an Eclipse XDB-C18 column (250 × 4.6 mm, 5 µm) (CH_3_CN/H_2_O = 65:35, flow rate 1.0 mL/min, λ = 254 nm), t_major_ = 16.027 min, t_minor_ = not found, dr > 99:1.

*Nickel(II)-(R)-N-(2-benzoyl-4-chlorophenyl)-1-(3,4-dichlorobenzyl)-2-methylpyrrolidine-2-carboxamide/(R)-2-(3-methoxyphenyl)glycine Schiff Base Complex* (**6j**). Red solid (99 mg, yield 70%); mp 135–137 °C; [*α*]^20^_D_ = –2183 (c = 1, CHCl_3_); ^1^H-NMR (400 MHz, CDCl_3_) *δ* 9.18 (d, *J* = 1.6 Hz, 1H), 8.23 (d, *J* = 9.3 Hz, 1H), 7.79 (d, *J* = 8.2 Hz, 1H), 7.54 (t, *J* = 7.6 Hz, 1H), 7.39 (dd, *J* = 15.2, 7.2 Hz, 3H), 7.29 (d, *J* = 7.5 Hz, 1H), 7.22 (t, *J* = 7.9 Hz, 1H), 7.16 (dd, *J* = 9.3, 2.5 Hz, 1H), 7.11–7.02 (m, 2H), 6.80 (dd, *J* = 8.2, 2.3 Hz, 1H), 6.66 (d, *J* = 2.5 Hz, 1H), 6.14 (d, *J* = 7.9 Hz, 1H), 4.71 (s, 1H), 4.08–3.95 (m, 1H), 3.80 (d, *J* = 13.0 Hz, 1H), 3.71 (s, 3H), 3.61 (d, *J* = 13.0 Hz, 1H), 3.34 (t, *J* = 8.9 Hz, 1H), 3.14 (dd, *J* = 18.8, 9.9 Hz, 1H), 2.46 (dd, *J* = 14.1, 9.9 Hz, 1H), 2.19 (dd, *J* = 17.6, 7.9 Hz, 2H), 1.49 (s, 3H). ^13^C-NMR (125 MHz, CDCl_3_) *δ* 180.80, 177.4, 171.7, 159.8, 141.3, 139.2, 135.8, 134.0, 133.3, 133.3, 133.1, 132.6, 132.4, 131.2, 130.0, 129.9, 129.7, 129.0, 128.9, 127.7, 127.1, 126.7, 126.0, 124.9, 118.4, 113.8, 112.6, 74.7, 74.4, 57.3, 55.4, 55.3, 41.7, 21.2, 18.3. LRMS (ESI, *m*/*z*): 720.0 [M + H]^+^. HRMS (ESI, *m*/*z*): calcd for C_35_H_31_Cl_3_N_3_NiO_4_, 720.0554 [M + H]^+^; found 720.0574. The dr was determined by LC-MS using an Eclipse XDB-C18 column (250 × 4.6 mm, 5 µm) (CH_3_CN/H_2_O = 65:35, flow rate 1.0 mL/min, λ = 254 nm), t_major_ = 16.851 min, t_minor_ = 18.454, dr = 98:2.

*Nickel(II)-(R)-N-(2-benzoyl-4-chlorophenyl)-1-(3,4-dichlorobenzyl)-2-methylpyrrolidine-2-carboxamide/(R)-2-(3-bromophenyl)glycine Schiff Base Complex* (**6k**). Red solid (134 mg, yield 87%); mp 102–104 °C; [*α*]^20^_D_ = –2069 (c = 1, CHCl_3_); ^1^H-NMR (400 MHz, CDCl_3_) *δ* 9.18 (d, *J* = 2.1 Hz, 1H), 8.22 (d, *J* = 9.3 Hz, 1H), 8.15 (s, 1H), 7.80 (dd, *J* = 8.2, 2.2 Hz, 1H), 7.57 (t, *J* = 7.6 Hz, 1H), 7.50–7.38 (m, 3H), 7.32 (dd, *J* = 18.2, 7.8 Hz, 2H), 7.20–7.08 (m, 3H), 6.67 (d, *J* = 2.5 Hz, 1H), 6.13 (d, *J* = 7.8 Hz, 1H), 4.71 (s, 1H), 4.04 (t, *J* = 10.9 Hz, 1H), 3.72 (dd, *J* = 78.9, 13.1 Hz, 2H), 3.41–3.29 (m, 1H), 3.14 (dd, *J* = 21.0, 13.1 Hz, 1H), 2.50 (dd, *J* = 13.7, 10.0 Hz, 1H), 2.30–2.04 (m, 2H), 1.50 (s, 3H), ^13^C-NMR (125 MHz, CDCl_3_) *δ* 180.8, 176.8, 172.1, 141.4, 139.9, 135.8, 134.0, 133.4, 132.9, 132.9, 132.4, 131.6, 131.3, 130.3, 130.2, 130.0, 129.3, 129.3, 129.1, 127.6, 127.0, 126.7, 126.1, 125.3, 125.0, 122.9, 74.5, 74.1, 57.3, 55.3, 41.8, 21.2, 18.3. LRMS (ESI, *m*/*z*): 767.9 [M + H]^+^. HRMS (ESI, *m*/*z*): calcd for C_34_H_28_BrCl_3_N_3_NiO_3_, 767.9557 [M + H]^+^; found 767.9561. The dr was determined by LC-MS using an Eclipse XDB-C18 column (250 × 4.6 mm, 5 µm) (CH_3_CN/H_2_O = 65:35, flow rate 1.0 mL/min, λ = 254 nm), t_major_ = 25.867 min, t_minor_ = 29.203, dr = 97:3.

*Nickel(II)-(R)-N-(2-benzoyl-4-chlorophenyl)-1-(3,4-dichlorobenzyl)-2-methylpyrrolidine-2-carboxamide/(R)-2-cyclobutylglycine Schiff Base Complex* (**6l**). Red solid (121 mg, yield 91%); mp 266–269 °C; [*α*]^20^_D_ = –2560 (c = 1, CHCl_3_); ^1^H-NMR (400 MHz, CDCl_3_) *δ* 9.14 (s, 1H), 8.24 (d, *J* = 9.3 Hz, 1H), 7.74–7.69 (m, 1H), 7.58–7.49 (m, 2H), 7.49–7.43 (m, 1H), 7.34–7.27 (m, 2H), 7.10 (dd, *J* = 9.3, 2.5 Hz, 1H), 6.88 (d, *J* = 7.4 Hz, 1H), 6.60 (d, *J* = 2.5 Hz, 1H), 4.21–4.05 (m, 1H), 3.84–3.75 (m, 2H), 3.52 (d, *J* = 13.0 Hz, 1H), 3.45–3.38 (m, 1H), 3.17–3.00 (m, 1H), 2.78–2.60 (m, 1H), 2.59–2.34 (m, 3H), 2.31–2.14 (m, 2H), 2.11–2.09 (m, 1H), 1.91–1.77 (m, 1H), 1.75–1.62 (m, 2H), 1.46 (s, 3H). ^13^C-NMR (125 MHz, CDCl_3_) *δ* 180.1, 177.0, 169.4, 140.7, 135.6, 133.7, 133.1, 132.9, 132.8, 132.3, 132.1, 130.9, 130.0, 129.6, 129.2, 128.9, 127.9, 127.8, 127.5, 125.6, 124.0, 74.3, 73.8, 57.0, 54.8, 41.7, 40.5, 25.8, 25.4, 20.5, 17.9, 17.3. LRMS (ESI, *m*/*z*): 666.0 [M − H]^−^. HRMS (ESI, *m*/*z*): calcd for C_32_H_29_Cl_3_N_3_NiO_3_, 666.0633 [M − H]^−^; found 666.0632. The dr was determined by LC-MS using Eclipse XDB-C18 column (250 × 4.6 mm, 5 µm) (CH_3_CN/H_2_O = 65:35, flow rate 1.0 mL/min, λ = 254 nm), t_major_ = 16.198 min, t_minor_ = not found, dr > 99:1.

*Nickel(II)-(R)-N-(2-benzoyl-4-chlorophenyl)-1-(3,4-dichlorobenzyl)-2-methylpyrrolidine-2-carboxamide/(R)-2-amino-4,4,4-trifluorobutyric acid Schiff Base Complex* (**6m**). Red solid (129 mg, yield 93%); mp 274–276 °C; [*α*]^20^_D_ = –3150 (c = 1, CHCl_3_); ^1^H-NMR (400 MHz, CDCl_3_) *δ* 9.16 (d, *J* = 1.8 Hz, 1H), 8.35 (d, *J* = 9.3 Hz, 1H), 7.68–7.47 (m, 4H), 7.36 (d, *J* = 8.1 Hz, 1H), 7.30 (d, *J* = 6.8 Hz, 1H), 7.16 (dd, *J* = 9.3, 2.5 Hz, 1H), 6.90 (d, *J* = 7.4 Hz, 1H), 6.60 (d, *J* = 2.5 Hz, 1H), 4.22–4.10 (m, 2H), 3.74–3.49 (m, 2H), 3.38 (t, *J* = 8.7 Hz, 1H), 3.09–2.80 (m, 1H), 2.63–2.46 (m, 1H), 2.42 (dd, *J* = 13.7, 9.8 Hz, 1H), 2.23–2.09 (m, 2H), 2.06–1.90 (m, 1H), 1.44 (s, 3H). ^13^C-NMR (125 MHz, CDCl_3_) *δ* 180.5, 177.2, 172.4, 141.9, 135.8, 134.0, 133.2, 133.2, 133.2, 133.1, 132.5, 131.3, 130.8, 129.9, 129.8, 129.7, 127.6, 127.6, 126.8, 125.8, 124.6, 74.9, 64.5, 57.8, 54.8, 41.2, 35.6 (d, *J* = 29.2 Hz), 20.3, 17.4. LRMS (ESI, *m*/*z*): 696.0 [M + H]^+^. HRMS (ESI, *m*/*z*): calcd for C_30_H_26_Cl_3_F_3_N_3_NiO_3_, 696.0164 [M + H]^+^; found 696.0171. The dr was determined by LC-MS using an Eclipse XDB-C18 column (250 × 4.6 mm, 5 µm) (CH_3_CN/H_2_O = 65:35, flow rate 1.0 mL/min, λ = 254 nm), t_major_ = 13.464 min, t_minor_ = not found, dr > 99:1.

*Nickel(II)-(R)-N-(2-benzoyl-4-chlorophenyl)-1-(3,4-dichlorobenzyl)-2-methylpyrrolidine-2-carboxamide/(R)-norvaline Schiff Base Complex* (**6n**). Red solid (118 mg, yield 90%); mp 227–228 °C; [*α*]^20^_D_ = –2430 (c = 1, CHCl_3_); ^1^H-NMR (400 MHz, CDCl_3_) *δ* 9.11 (d, *J* = 2.0 Hz, 1H), 8.21 (d, *J* = 9.3 Hz, 1H), 7.73 (dd, *J* = 8.2, 2.0 Hz, 1H), 7.61–7.49 (m, 2H), 7.46 (t, *J* = 7.4 Hz, 1H), 7.31 (dd, *J* = 15.9, 7.6 Hz, 2H), 7.12 (dd, *J* = 9.3, 2.6 Hz, 1H), 6.86 (d, *J* = 7.5 Hz, 1H), 6.60 (d, *J* = 2.6 Hz, 1H), 4.11–3.98 (m, 1H), 3.87 (dd, *J* = 7.6, 3.6 Hz, 1H), 3.65 (dd, *J* = 82.3, 13.0 Hz, 2H), 3.38 (t, *J* = 8.8 Hz, 1H), 3.25–3.02 (m, 1H), 2.48–2.37 (m, 1H), 2.27–2.14 (m, 2H), 2.13–1.97 (m, 1H), 1.84 (dt, *J* = 13.1, 8.4 Hz, 1H), 1.69 (dd, *J* = 13.1, 6.3 Hz, 1H), 1.59–1.49 (m, 1H), 1.45 (s, 3H), 0.79 (t, *J* = 7.2 Hz, 3H). ^13^C-NMR (125 MHz, CDCl_3_) *δ* 180.4, 179.0, 169.8, 140.9, 135.8, 133.9, 133.3, 133.1, 133.1, 132.4, 131.2, 130.3, 129.9, 129.5, 129.3, 128.1, 127.5, 127.5, 125.9, 124.5, 74.5, 70.5, 57.2, 55.0, 41.6, 37.4, 21.0, 18.4, 17.9, 13.9. LRMS (ESI, *m*/*z*): 656.1 [M + H]^+^. HRMS (ESI, *m*/*z*): calcd for C_31_H_31_Cl_3_N_3_NiO_3_, 656.0604 [M + H]^+^; found 656.0601. The dr was determined by LC-MS using an Eclipse XDB-C18 column (250 × 4.6 mm, 5 µm) (CH_3_CN/H_2_O = 65:35, flow rate 1.0 mL/min, λ = 254 nm), t_major_ = 14.850 min, t_minor_ = 17.260, dr = 98:2.

*Nickel(II)-(R)-N-(2-benzoyl-4-chlorophenyl)-1-(3,4-dichlorobenzyl)-2-methylpyrrolidine-2-carboxamide/(R)-valine Schiff Base Complex* (**6o**). Red solid (126 mg, yield 96%); mp 224–226 °C; [*α*]^20^_D_ = −2566 (c = 1, CHCl_3_); ^1^H-NMR (400 MHz, CDCl_3_) *δ* 9.18 (d, *J* = 2.1 Hz, 1H), 8.29 (d, *J* = 9.3 Hz, 1H), 7.68–7.64 (m, 1H), 7.60–7.50 (m, 2H), 7.49–7.43 (m, 1H), 7.35–7.30 (m, 2H), 7.12 (dd, *J* = 9.3, 2.6 Hz, 1H), 6.88–6.84 (m, 1H), 6.62 (d, *J* = 2.6 Hz, 1H), 4.13–4.01 (m, 1H), 3.82–3.76 (m, 2H), 3.58–3.52 (m, 1H), 3.45–3.38 (m, 1H), 3.07–2.87 (m, 1H), 2.49–2.39 (m, 1H), 2.25–2.10 (m, 2H), 1.86 (d, *J* = 6.6 Hz, 3H), 1.81–1.70 (m, 1H), 1.46 (s, 3H), 0.75 (d, *J* = 6.8 Hz, 3H). ^13^C-NMR (150 MHz, CDCl_3_) *δ* 180.2, 177.1, 170.1, 140.8, 135.7, 133.8, 133.0, 133.0, 132.8, 132.3, 132.2, 130.9, 130.0, 129.6, 129.3, 128.9, 127.7, 127.6, 127.3, 125.6, 124.0, 75.4, 74.2, 57.1, 54.7, 41.5, 34.2, 20.3, 19.7, 17.6, 17.5. LRMS (ESI, *m*/*z*): 656.0 [M − H]^−^. HRMS (ESI, *m*/*z*): calcd for C_31_H_29_Cl_3_N_3_NiO_3_, 656.0633 [M − H]^−^; found 656.0629. The dr was determined by LC-MS using an Eclipse XDB-C18 column (250 × 4.6 mm, 5 µm) (CH_3_CN/H_2_O = 65:35, flow rate 1.0 mL/min, λ = 254 nm), t_major_ = 14.706 min, t_minor_ = not found, dr > 99:1.

*Nickel(II)-(R)-N-(2-benzoyl-4-chlorophenyl)-1-(3,4-dichlorobenzyl)-2-methylpyrrolidine-2-carboxamide/(R)-leucine Schiff Base Complex* (**6p**). Red solid (127 mg, yield 95%); mp 126–127 °C; [*α*]^20^_D_ = −1898 (c = 1, CHCl_3_); ^1^H-NMR (500 MHz, CDCl_3_) *δ* 9.11 (d, *J* = 2.0 Hz, 1H), 8.15 (d, *J* = 9.3 Hz, 1H), 7.78 (dd, *J* = 8.2, 2.1 Hz, 1H), 7.60–7.49 (m, 2H), 7.46 (td, *J* = 7.5, 1.0 Hz, 1H), 7.32 (t, *J* = 6.8 Hz, 2H), 7.10 (dd, *J* = 9.3, 2.6 Hz, 1H), 6.87 (d, *J* = 7.6 Hz, 1H), 6.60 (d, *J* = 2.6 Hz, 1H), 4.06–3.98 (m, 1H), 3.83 (dd, *J* = 10.5, 4.1 Hz, 1H), 3.80 (d, *J* = 13.2 Hz, 1H), 3.53 (d, *J* = 13.1 Hz, 1H), 3.39 (t, *J* = 8.8 Hz, 1H), 3.32–3.15 (m, 1H), 2.49–2.31 (m, 2H), 2.29–2.16 (m, 2H), 1.99–1.81 (m, 1H), 1.45 (s, 3H), 1.38–1.32 (m, 1H), 0.81 (d, *J* = 6.7 Hz, 3H), 0.38 (d, *J* = 6.5 Hz, 3H). ^13^C-NMR (125 MHz, CDCl_3_) *δ* 180.4, 178.8, 169.2, 140.8, 135.8, 133.9, 133.3, 133.1, 132.9, 132.2, 131.1, 130.3, 129.9, 129.4, 129.3, 128.1, 127.8, 127.6, 125.9, 124.5, 74.4, 69.3, 56.9, 55.3, 46.2, 41.8, 24.5, 23.9, 21.4, 21.0, 18.1. LRMS (ESI, *m*/*z*): 669.9 [M + H]^+^. HRMS (ESI, *m*/*z*): calcd for C_32_H_33_Cl_3_N_3_NiO_3_, 670.0935 [M + H]^+^; found 670.0935. The dr was determined by LC-MS using an Eclipse XDB-C18 column (250 × 4.6 mm, 5 µm) (CH_3_CN/H_2_O = 65:35, flow rate 1.0 mL/min, λ = 254 nm), t_major_ = 19.979 min, t_minor_ = not found, dr > 99:1.

*Nickel(II)-(R)-N-(2-benzoyl-4-chlorophenyl)-1-(3,4-dichlorobenzyl)-2-methylpyrrolidine-2-carboxamide/(R)-methionine Schiff Base Complex* (**6q**). Red solid (120 mg, yield 88%); mp 123–125 °C; [*α*]^20^_D_ = −2202 (c = 1, CHCl_3_); ^1^H-NMR (400 MHz, CDCl_3_) *δ* 9.11 (d, *J* = 1.5 Hz, 1H), 8.21 (d, *J* = 9.3 Hz, 1H), 7.72 (d, *J* = 6.7 Hz, 1H), 7.60–7.51 (m, 2H), 7.47 (t, *J* = 7.4 Hz, 1H), 7.32 (dd, *J* = 14.6, 7.6 Hz, 2H), 7.13 (dd, *J* = 9.3, 2.5 Hz, 1H), 6.88 (d, *J* = 7.5 Hz, 1H), 6.60 (d, *J* = 2.5 Hz, 1H), 4.09–3.96 (m, 1H), 3.93 (dd, *J* = 8.3, 3.6 Hz, 1H), 3.66 (dd, *J* = 74.8, 13.0 Hz, 2H), 3.37 (t, *J* = 8.9 Hz, 1H), 3.17 (dd, *J* = 16.5, 10.3 Hz, 1H), 3.09–3.01 (m, 1H), 2.67–2.53 (m, 1H), 2.49–2.37 (m, 1H), 2.19 (dd, *J* = 15.7, 6.7 Hz, 3H), 1.97 (s, 3H), 1.91–1.77 (m, 1H), 1.45 (s, 3H). ^13^C-NMR (125 MHz, CDCl_3_) *δ* 180.4, 178.3, 170.3, 141.1, 135.7, 133.9, 133.3, 133.2, 132.9, 132.6, 132.4, 131.2, 130.4, 129.9, 129.6, 129.4, 127.9, 127.5, 127.4, 125.9, 124.6, 74.5, 69.8, 57.3, 55.0, 41.6, 35.1, 29.8, 21.3, 18.0, 15.8. LRMS (ESI, *m*/*z*): 688.0 [M − H]^−^. HRMS (ESI, *m*/*z*): calcd for C_31_H_31_Cl_3_N_3_NiO_3_S, 688.0324 [M − H]^−^; found 688.0339. The dr was determined by LC-MS using an Eclipse XDB-C18 column (250 × 4.6 mm, 5 µm) (CH_3_CN/H_2_O = 65:35, flow rate 1.0 mL/min, λ = 254 nm), t_major_ = 13.870 min, t_minor_ = 14.997, dr = 98:2.

*(S)-Phenylalanine* (**5a**). White solid (16.5 mg, yield 88%); mp 273–275 °C; [*α*]^20^_D_ = – 38 (c = 1, MeOH); ^1^H-NMR (500 MHz, D_2_O) *δ* 7.41 (m, *J* = 14.6, 7.2 Hz, 3H), 7.33 (d, *J* = 7.1 Hz, 2H), 4.00 (dd, *J* = 7.9, 5.2 Hz, 1H), 3.30 (dd, *J* = 14.5, 5.2 Hz, 1H), 3.13 (dd, *J* = 14.5, 8.0 Hz, 1H), 2.00 (s, 1H), 1.96 (s, 2H). ^13^C-NMR (125 MHz, D_2_O) *δ* 173.4, 134.6, 128.8, 128.6, 127.2, 55.5, 35.8, 22.1, 20.7. LRMS (ESI, *m/z*): 166.0 [M + H]^+^. HRMS (ESI, *m/z*): calcd for C_9_H_11_NO_2_, 166.0863 [M + H]^+^; found 166.0865. The ee was determined by HPLC using an Astec CHIROBIOTIC™ T chiral HPLC column (4.6 mm × 25 cm, 5 μm) (MeOH/H_2_O = 90/10, λ = 210 nm, 1 mL/min). t_S_ = 7.472 min, t_R_ = not found, ee > 99 %.

X-ray Single Crystal Structure Analysis of (*S*,2*S*)-**6a**: (*S*,2*S*)-**6a** (100 mg) was dissolved in DCM (10 mL) and crystals suitable for X-ray single crystal structure determination were obtained after all the solvent had evaporated at room temperature. The X-ray structure of (*S*,2*S*)**-6a** was resolved at T = 293(2) K: C_3__5_H_30_Cl_3_N_3_NiO_3_, *Mr* = 705.68, monoclinic. Space group *P2* (1), a = 10.1494 (12) Å, b = 15.7644 (19) Å, c = 22.968 (3) Å, α = 90°, β = 90°, γ = 90°, *V* = 3674.9 (8) Å^3^, *Z* = 4. These data can be obtained free of charge from the Cambridge Crystallographic Data Centre via www.ccdc.cam.ac.uk/data_request/cif, the CCDC number is 1472366.

## 4. Conclusions

In conclusion, we have developed a generalized method for DKR of *C,N*-unprotected racemic *α*-amino acids with a novel, stable, and recyclable *α*-methylproline-derived chiral ligand under operationally convenient conditions. Compared to the reported methods, we realized good to excellent yields and diastereoselectivities of phenyl-substituted glycines. In addition, the substrates scope range includes compounds such as cyclobutyl- and trifluoroethyl-substituted glycine, also with excellent yields and diastereoselectivities. This new approach offers versatile access to various chiral *α*-amino acids in high yields (up to 98%) with excellent ee values (ee >99%). Particularly noteworthy is that the novel designed *α*-methylproline-derived chiral ligand is more stable than the previous reported proline-derived chiral ligand.

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
