# Peer review of "Recyclable and Stable α-Methylproline-Derived Chiral Ligands for the Chemical Dynamic Kinetic Resolution of free C,N-Unprotected α-Amino Acids"

_molecules, 2019, doi:10.3390/molecules24122218_

Reviewer 1 Report

This is a useful extension of previous work and could find use, even though it is not an especially practical method - not many are in this area. However, at least the basic experimental demands are simple, but the requirement for chromatography rather detracts from the practicality.

The full descriptions are supported by excellent characterisation data and decent diagrams - one point in this respect however - could the authors redraw the key structures 4 and 5 to more clearly show the central stereogenic centre?

Otherwise a decent contribution, which could lead to better methods.

Author Response

Comments:

This is a useful extension of previous work and could find use, even though it is not an especially practical method - not many are in this area. However, at least the basic experimental demands are simple, but the requirement for chromatography rather detracts from the practicality.

The full descriptions are supported by excellent characterisation data and decent diagrams - one point in this respect however - could the authors redraw the key structures 4 and 5 to more clearly show the central stereogenic centre? Otherwise a decent contribution could lead to better methods.

Question 1: Could the authors redraw the key structures 4 and 5 to more clearly show the central stereogenic centre?

Response: We are very grateful to the reviewer’s comments.

We have redrawn the structures 4 and 5 in the revised manuscript.

Reviewer 2 Report

This paper reports on a new chiral α-methylproline derivative capable to perform resolution of C,N-unprotected racemic α-amino acid. The resolution process appears to give high yields and diastereoselectivities, in particular in the case of phenyl-substituted glycines.

Chiral resolution of amino acids is a subject of undoubtedly large interest, considering the biological relevance of these compounds and their potential pharmacological applications. The authors report a new compound able to efficiently perform this task and, therefore, I suggest publication of this paper.   

I have just a revision to suggest.  The crystal structure of compound 6a is of relevance in the context of this Ms and it should reported in the main text of the paper rather than as supplementary information, together with a short experimental part, including the method used to obtain the crystals and to solve the X-ray structure.

Author Response

Comments:

This paper reports on a new chiral α-methylproline derivative capable to perform resolution of C,N-unprotected racemic α-amino acid. The resolution process appears to give high yields and diastereoselectivities, in particular in the case of phenyl-substituted glycines.

Chiral resolution of amino acids is a subject of undoubtedly large interest, considering the biological relevance of these compounds and their potential pharmacological applications. The authors report a new compound able to efficiently perform this task and, therefore, I suggest publication of this paper.   

I have just a revision to suggest.  The crystal structure of compound 6a is of relevance in the context of this Ms and it should be reported in the main text of the paper rather than as supplementary information, together with a short experimental part, including the method used to obtain the crystals and to solve the X-ray structure.

Question 1: The crystal structure of compound 6a should be reported in the main text, together with a short experimental part, including the method used to obtain the crystals and to solve the X-ray structure.

Response: We are very grateful to the reviewer’s comments.

We have added compound 6a and the method used to obtain the crystals and to solve the X-ray structure in the main text in the revised manuscript.